# A unifying account of replay as context-driven memory reactivation

**Zhenglong Zhou\*, Michael J Kahana[†], Anna C Schapiro[†]**

Department of Psychology, University of Pennsylvania, Philadelphia, United States

## eLife Assessment

This is an **important** account of replay as recency-weighted context-guided memory reactivation that explains a number of empirical findings across human and rodent memory literatures. The evidence is **compelling** and the work is likely to inspire further adaptions to incorporate additional biological and cognitive features.

**Abstract** During rest and sleep, sequential neural activation patterns corresponding to awake experience re-emerge, and this replay has been shown to benefit subsequent behavior and memory. Whereas some studies show that replay directly recapitulates recent experience, others demonstrate that replay systematically deviates from the temporal structure, the statistics, and even the content of recent experience. Given these disparate characteristics, what is the nature and purpose of replay? Here, we offer a theoretical framework in which replay reflects simple context-guided processes that facilitate memory. We suggest that during awake learning, the brain associates experiences with the contexts in which they are encoded, at encoding rates that vary according to the salience of each experience. During quiescence, replay emerges as the result of a cascade of autonomous bidirectional interactions between contexts and their associated experiences, which in turn facilitates memory consolidation. A computational model instantiating this proposal explains numerous replay phenomena, including findings that existing models fail to account for and observations that have been predominantly construed through the lens of reinforcement learning. Our theory provides a unified, mechanistic framework of how the brain initially encodes and subsequently replays experiences in the service of memory consolidation.

**\*For correspondence:**
zzhou34@sas.upenn.edu

[†]These authors contributed equally to this work

**Competing interest:** The authors declare that no competing interests exist.

## Introduction

Sleep and rest are crucial for learning and memory (*Klinzing et al., 2019*; *Walker and Stickgold, 2004*; *Wamsley, 2019*). A candidate mechanism facilitating these benefits is replay – the offline re-emergence of neural activity associated with awake experience. Over the past several decades, the field of neuroscience has accumulated extensive evidence of replay (*Foster, 2017*; *Liu et al., 2022*; *Ólafsdóttir et al., 2018*; *Sheridan et al., 2024*) and increasing evidence of its utility to behavior (*Dupret et al., 2010*; *Ego-Stengel and Wilson, 2010*; *Liu et al., 2021*; *Girardeau et al., 2009*; *Jadhav et al., 2012*). It is challenging, however, to characterize the principles and functions of replay, because it exhibits disparate characteristics across states and task contexts that are difficult to synthesize under one framework. Early studies showed that replay preserves the correlational structure and the temporal structure of multi-cell spiking patterns that underlie awake experiences (*Wilson and McNaughton, 1994*; *Skaggs and McNaughton, 1996*; *Nádasdy et al., 1999*). Most canonically, the firing of sequences of hippocampal place cells corresponding to traversals through an environment re-emerges with the same sequential firing during rest and sleep. Subsequent studies, however, demonstrated that replay deviates from the temporal structure, statistics, and content of

recent experience in myriad ways: Replay activates never-experienced novel trajectories (*Gupta et al., 2010*), over-represents salient experiences (*Singer and Frank, 2009*; *Ólafsdóttir et al., 2015*; *Wu et al., 2017*), unrolls in the reverse order of a behavioral sequence when an animal consumes reward (*Ambrose et al., 2016*; *Foster and Wilson, 2006*; *Liu et al., 2019*; *Michon et al., 2019*), and sometimes exhibits a bias away from recently experienced trajectories (*Carey et al., 2019*; *Gupta et al., 2010*). These observations illustrate that replay is not a simple, direct recapitulation of awake experience.

An influential explanation for why replay exhibits distinctive properties, especially in the presence of reward, is that replay is for learning value predictions in the manner of reinforcement learning (RL) models (*Ambrose et al., 2016*; *Foster and Wilson, 2006*; *Liu et al., 2021*; *Mattar and Daw, 2018*; *Momennejad et al., 2018*). According to this perspective, adaptive behavior depends on identifying actions that lead to rewarding outcomes, which requires predicting the downstream value of actions (i.e. eventual punishment or reward). This perspective argues that replay reactivates memories to update these predictions. For example, many speculate that reverse replay implements a classic method for updating value predictions, which is to propagate information backward from a reward through experienced trajectories to update upstream actions' value predictions (*Sutton, 1988*). A recent theory extending this perspective (*Mattar and Daw, 2018*) argues that a range of other replay characteristics (*Ambrose et al., 2016*; *Cheng and Frank, 2008*; *Diba and Buzsáki, 2007*; *Wu et al., 2017*; *Ólafsdóttir et al., 2015*) can be explained by assuming that replay prioritizes updates that will most improve future behavior. The model assumes that replay knows in advance which updates will best improve behavior, defined by a quantity called the expected value of backup (EVB). When replay progresses from updates that improve future behavior the most to the least (i.e. highest to lowest EVB), it produces patterns that match a number of empirically observed phenomena. However, knowing the behavioral consequence of replay before it is performed is implausible, and the model makes predictions that are inconsistent with empirical data, including the prediction that replay always prioritizes updates relevant to the present goal (*Carey et al., 2019*; *Gillespie et al., 2021*; *Gupta et al., 2010*) and that learning reduces the rate of backward more than forward replay (*Igata et al., 2021*; *Shin et al., 2019*). Thus, this RL perspective is unlikely to provide a complete account of the characteristics of replay.

We offer an alternative theoretical account in which replay reflects simple context-guided memory processes. We hypothesize that, during awake learning, the brain sequentially associates experiences with the contexts in which they are encoded, in a manner modulated by the salience of each experience (more rapid association for more salient experiences). During quiescence – both awake rest periods and sleep, replay arises as the result of a cascade of bidirectional interactions between contexts and their associated experiences. The offline brain continues to learn from this replay, updating associations between reactivated memories and contexts. In this account, replay does not compute the utility of memories for learning value predictions, nor does it track value predictions. Instead, replay arises naturally from a memory-based mechanism operating bidirectionally between contexts and their associated experiences.

We show that an instantiation of this account – a computational model that builds on established context-based memory encoding and retrieval mechanisms (*Howard and Kahana, 2002*; *Polyn et al., 2009*) – unifies numerous replay phenomena. These include replay patterns often presumed to involve RL computations and findings that existing models do not account for. We focus on the model as a formulation of hippocampal replay, capturing how the hippocampus may replay past experiences through simple and interpretable mechanisms. First, the content and structure of replay sequences in the model vary according to states and task contexts in ways that mirror empirical observations (*Bendor and Wilson, 2012*; *Diba and Buzsáki, 2007*; *Foster and Wilson, 2006*; *Wikenheiser and Redish, 2013*). Second, the model captures prominent effects of reward on replay (*Ambrose et al., 2016*; *Singer and Frank, 2009*; *Ólafsdóttir et al., 2015*) despite not tracking value predictions. Third, in line with a number of findings (*Barron et al., 2020*; *Gupta et al., 2010*; *Karlsson and Frank, 2009*; *Liu et al., 2019*; *Wimmer et al., 2023*), replay is not restricted to direct recent experience: The model reactivates non-local and never-experienced novel trajectories. Moreover, the model captures a range of experience-dependent replay characteristics (*Berners-Lee et al., 2022*; *Carey et al., 2019*; *Cheng and Frank, 2008*; *Diba and Buzsáki, 2007*; *Gupta et al., 2010*; *Igata et al., 2021*; *Karlsson and Frank, 2009*; *Shin et al., 2019*). Finally, replay benefits memory consolidation in ways that align

with prior observations and theories (*Born and Wilhelm, 2012*; *Carr et al., 2011*; *King et al., 2017*; *Liu et al., 2021*; *Michon et al., 2019*; *Payne and Kensinger, 2010*; *Payne et al., 2008*). As a whole, our framework provides a general, mechanistic account of how the hippocampus initially encodes and subsequently reactivates experiences in the service of memory consolidation.

## Results

### A context model of memory replay

Our proposed model builds on memory encoding and retrieval mechanisms established in retrieved-context models of memory, as exemplified in the context maintenance and retrieval model (CMR; *Howard and Kahana, 2002*; *Polyn et al., 2009*). We refer to the model as CMR-replay (*Figure 1b*). We begin with an overview of the model architecture, followed by illustrations of awake learning and replay in the model by considering how it encodes and reactivates sequences of items (*Figure 1a*). We provide a detailed walk-through of operations in the model in Methods.

CMR-replay comprises four elements (*Figure 1b*): item ($f$), context ($c$), item-to-context associations ($M^{fc}$), and context-to-item associations ($M^{cf}$). The values of the elements of the $f$ and $c$ vectors can be considered to correspond abstractly to the firing rate of a neuron or population of neurons, and the values of $M^{cf}$ and $M^{fc}$ to the strength of synaptic connections between those neurons. Prior to learning, distinct items ($f$) are associated with orthogonal context features. As CMR-replay encodes (during awake learning) or reactivates (during replay) a sequence of items, $f$ represents the current item, whereas a drifting context ($c$) maintains a receding history of present and past items' associated contexts (*Figure 1c*), which resets before the model encodes each sequence. We represent the drifting context during learning and replay with $c$ and an item's associated context with $c_f$. During both awake learning and replay, CMR-replay associates each item with the current drifting context by updating bidirectional associations between them – item-to-context associations ($M^{fc}$), which map each item to its associated context, and context-to-item associations ($M^{cf}$), which map each context to associated items. Prior to learning, these associations are fully orthogonal: $M^{fc}$ maps each item to a distinct context and $M^{cf}$ maps each context feature to a distinct item.

During awake encoding of a sequence of items, for each item $f$, the model retrieves its associated context $c_f$ via $M^{fc}$. The drifting context $c$ incorporates the item's associated context $c_f$ and downweights its representation of previous items' associated contexts (*Figure 1c*). Thus, the context layer maintains a recency-weighted sum of past and present items' associated contexts. To perform encoding, CMR-replay updates $M^{fc}$ and $M^{cf}$ to strengthen associations between the current item and context (*Figure 1d*). The $M^{fc}$ update adds the current context to the context associated with the item. Because the current context contains contexts associated with previous items, through the $M^{fc}$ update, the context associated with the item starts to reflect contexts of prior items. For the same reason, through the $M^{cf}$ update, $M^{cf}$ learns to map contexts associated with previous items to the current item.

Building on prior work (*Cohen and Kahana, 2022*; *Talmi et al., 2019*), CMR-replay embraces the simplifying assumption that the salience of each item influences its rate of encoding (i.e. the learning rates at which the model updates $M^{fc}$ and $M^{cf}$). In particular, the model updates $M^{fc}$ and $M^{cf}$ at faster rates for salient items, including those that are novel and rewarding (see Methods), than for others. Higher encoding rates allow salient items to form associations with their encoding contexts more rapidly. In CMR-replay, salience modifies encoding rates only for the current item and context; it does not modify encoding rates for the items that lead up to the salient item.

During replay, the model generates a cascade of item and context reactivations by operating bidirectionally on $M^{fc}$ and $M^{cf}$. This process begins with an initial item reactivation. At the onset of each replay period, the model selects this item from an activity distribution that reflects spontaneous activity during awake rest or sleep (*Figure 1e*). To capture the near absence of external sensory information during sleep, we initialized this distribution with random activity across all items at sleep onset. In simulations of awake rest, in contrast, we present a cue that represents the external sensory context of where the model is 'resting'. This cue evokes activities that bias the distribution toward this resting location. As a result, awake replay exhibits an initiation bias – a tendency to initiate at the item most strongly associated with the cue.

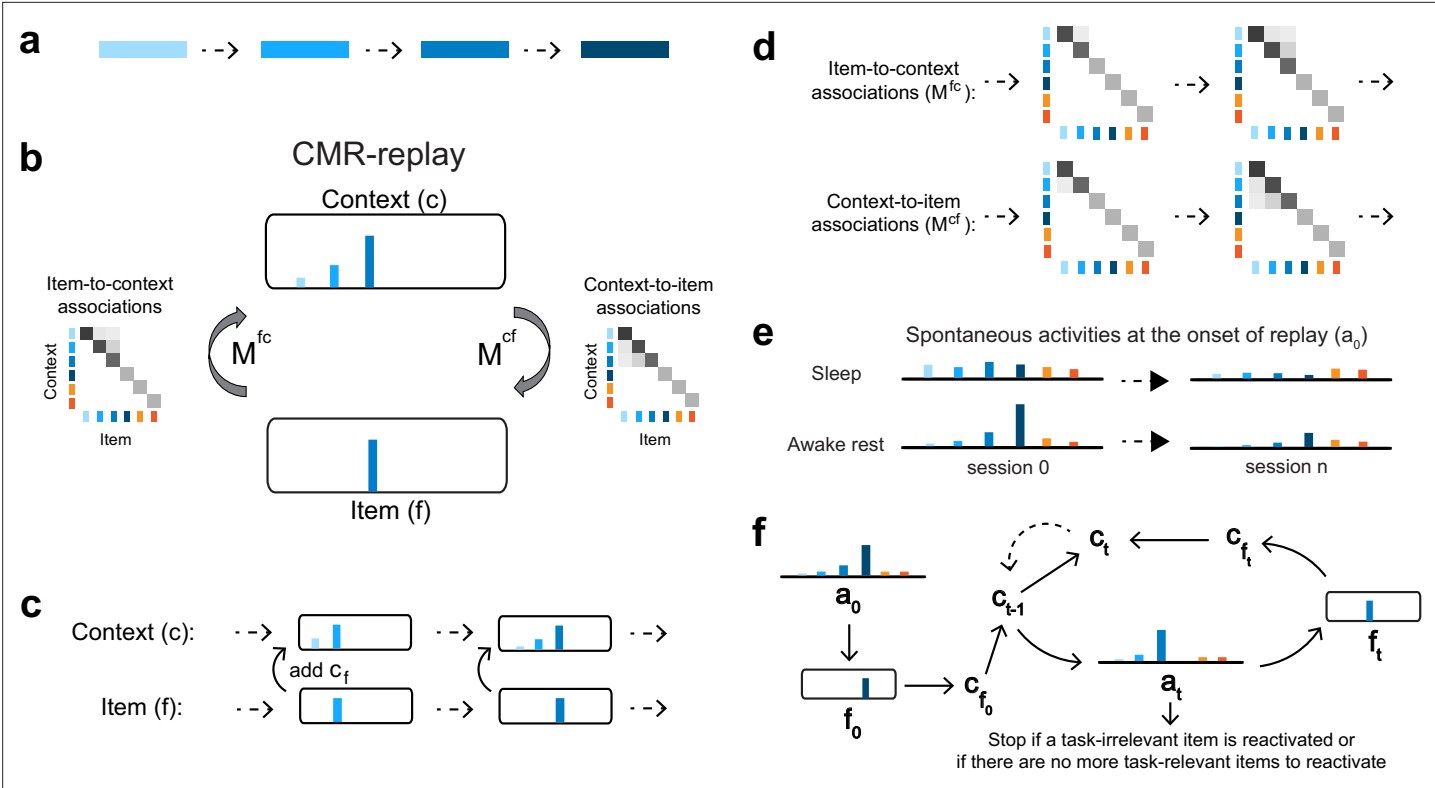

**Figure 1.** CMR-replay. (**a**) Consider a task of encoding a sequence consisting of four items, each denoted by a shade of blue. (**b**) We propose a model of replay that builds on the CMR, which we refer to as CMR-replay. The model consists of four components: item ($f$), context ($c$), item-to-context associations ($M^{fc}$), and context-to-item associations ($M^{cf}$). At each timestep during awake encoding, $f$ represents the current item and $c$ is a recency-weighted average of associated contexts of past and present items. CMR-replay associates $f$ and $c$ at each timestep, updating $M^{fc}$ and $M^{cf}$ according to a Hebbian learning rule. $M^{fc}$ and $M^{cf}$, respectively, support the retrieval of an item's associated context and a context's associated items. During replay, $f$ represents the current reactivated item, and $c$ is a drifting context representing a recency-weighted average of associated contexts of past and present reactivated items. Here, too, the model updates $M^{fc}$ and $M^{cf}$ to associate reactivated $f$ and $c$. The figure illustrates the representations of $f$, $c$, $M^{fc}$, and $M^{cf}$ as the model encodes the third item during learning. Lengths of color bars in $f$ and $c$ represent relative magnitudes of different features. Shades of gray illustrate the weights in $M^{fc}$ and $M^{cf}$. Orange features represent task-irrelevant items, which do not appear as inputs during awake encoding but compete with task-relevant items for reactivation during replay. (**c**) During both awake encoding and replay, context $c$ drifts by incorporating the current item $f$'s associated context $c_f$ and downweighting previous items' associated contexts. The figure illustrates how context drifts during the first time the model encodes the example sequence. (**d**) The figure illustrates $M^{fc}$ and $M^{cf}$ updates as the model encodes the third item during the first presentation of the sequence. (**e**) Consider the activation of items at the onset of sleep and awake rest across sessions of learning. At replay onset, an initial probability distribution across items $a_0$ varies according to the behavioral state (i.e. awake, rest, or sleep). Compared to sleep, during awake rest, $a_0$ is strongly biased toward features associated with external inputs during awake rest. For awake rest, the figure shows an example of $a_0$ when the model receives a context cue related to the fourth item. Through repeated exposure to the same task sequence across sessions of learning, activities of the four task-related items (i.e. blue items) become suppressed in $a_0$ relative to task-irrelevant items (i.e. orange items). (**f**) Each replay period begins by sampling an item $f_{t=0}$ according to $a_0$, where $t$ denotes the current timestep. If $f_{t=0}$ is a task-related item, its associated context $c_{f_{t=0}}$ is reinstated as $c_0$ to enter a recursive process. During this process, at each timestep $t \geq 1$, $c_{t-1}$ evokes a probability distribution $a_t$ that excludes previously reactivated items. Given $a_t$, the model samples an item $f_t$ and reinstates $f_t$'s associated context $c_{f_t}$, which is combined with $c_{t-1}$ to form a new context $c_t$ to guide the ensuing reactivation. The dashed arrow indicates that $c_t$ becomes $c_{t-1}$ for the next timestep. At any $t$, the replay period ends with a probability of 0.1 or if a task-irrelevant item is reactivated.

Once CMR-replay reactivates an initial item, this triggers a sequence of autonomous reactivation (*Figure 1f*). As in awake encoding, during replay, the model maintains a drifting context – a recency-weighted average of past and present reactivated items' associated contexts. At each timestep, using the current context as a cue, the model evokes a distribution of activities across items via $M^{cf}$. The model converts these activities into a probability distribution and samples another item without replacement (i.e. by excluding previously reactivated items). This mechanism of sampling without replacement, akin to response suppression in established context memory models (*Howard and Kahana, 2002*), could be implemented by neuronal fatigue or refractory

dynamics (*Burgess, 1992*; *Grossberg, 1978*). Non-repetition during reactivation is also a common assumption in replay models that regulate reactivation through inhibition or prioritization (*Diekmann and Cheng, 2023*; *Mattar and Daw, 2018*; *Singh et al., 2022*). In accordance with awake encoding, the model updates context by incorporating the newly reactivated item's associated context and updates $M^{fc}$ and $M^{cf}$ to associate the reactivated item with the updated context, albeit at much slower rates. The updated context then guides item reactivation at the next timestep. At each timestep, a replay event ends with a constant probability or if a task-irrelevant item becomes reactivated.

In the model, replay tends to preserve the temporal contiguity of awake experience, such that each reactivated item tends to be followed by the item that was encoded immediately after or before it (Figure 6d, left). During awake encoding, because of the way context incrementally drifts, the encoding contexts for adjacent items are more similar than for items that are far apart, except when a distractor intervenes between two adjacent items to demarcate an event boundary. During replay, when the model retrieves a reactivated item's associated context to guide the next reactivation, it will then favor the reactivation of items that immediately preceded or followed the current item during awake encoding (Figure 6d, left). This behavior is referred to as the model's contiguity bias, which allows replay to generate coherent sequences despite its stochasticity. Its contiguity bias stems from its use of shared, temporally autocorrelated context to link successive items, despite the orthogonal nature of individual item representations. This bias would be even stronger if items had overlapping representations, as observed in place fields. Following prior work (*Mattar and Daw, 2018*), we consider replay events to be replayed sequences (one per replay period) with consecutive segments of length five or greater that preserve the contiguity of the awake sequence.

In memory models that reactivate memories for offline learning (*Ans and Rousset, 2000*; *Meeter, 2003*; *Norman et al., 2005*), a common issue is that the most well-learned items are rehearsed most often, leading to additional strengthening of these items, leading in turn to even more rehearsal, and so on. CMRs can exhibit this same rich-get-richer phenomenon. The solution in prior models has been to incorporate a mechanism that balances rehearsal across items (*Meeter, 2003*; *Norman et al., 2005*; *Singh et al., 2022*). CMR-replay has such a mechanism as well, which increasingly downweights task-related items at the onset of replay through repetition in the same task. This process downweights items according to the activity of their retrieved contexts in the preceding awake encoding period. As CMR-replay repeatedly strengthens weights for a sequence of items, the probability that replay begins with its constituent items decreases, allowing alternative items that received less exposure to participate in replay (*Figure 1e*). The proposal that a suppression mechanism plays a role in replay aligns with models that regulate place cell reactivation via inhibition (*Malerba et al., 2016*) and with empirical observations of increased hippocampal inhibitory interneuron activity with experience (*Berners-Lee et al., 2022*). Our model assumes the presence of such inhibitory mechanisms but does not explicitly model them. There are multiple possibilities for how a biological process may implement something like our suppression mechanism (*Hasselmo et al., 1996*). Though more empirical investigation is needed to determine the implementation, the need across theoretical perspectives for some form of balancing mechanism motivates the strong prediction that some biological mechanism like this must be at work.

We simulate awake learning in a number of tasks (*Ambrose et al., 2016*; *Bendor and Wilson, 2012*; *Carey et al., 2019*; *Diba and Buzsáki, 2007*; *Gupta et al., 2010*; *Liu et al., 2019*; *Liu et al., 2021*; *Ólafsdóttir et al., 2015*) by exposing the model to sequences of items that correspond to trajectories of spatial locations or visual stimuli as experienced by animals in the experiments. In between sessions of wake learning, we simulate quiescence (both awake rest and sleep) as periods of autonomous reactivation. Our objective is to examine whether CMR-replay can capture qualitative aspects of existing replay phenomena, rather than to provide a quantitative fit to the data. Unlike prior work that finds different best-fitting parameters across simulations (*Polyn et al., 2009*; *Lohnas et al., 2015*; *Cohen and Kahana, 2022*), CMR-replay employs one set of model parameters across all simulations. In the following sections, we show that the model accounts for a diverse range of empirical phenomena, including context-dependent variations of replay (Figure 2), effects of reward on replay (Figure 3), replay patterns that go beyond direct recent experience (Figure 4), experience-dependent variations of replay (Figure 5), and ways in which replay facilitates memory (Figure 6).

## The context dependency of memory replay

During quiescence, sequential neural firing during sharp-wave ripples (SWRs) recapitulates the temporal pattern of previous waking experience (*Foster, 2017*). We distinguish between forward and backward replay, defined as neural activity that either preserves the order of a prior experience (forward replay) or reverses it (backward replay). In animals and humans, the content and directionality of replay systematically vary according to task contexts and behavioral states (*Bendor and Wilson, 2012*; *Diba and Buzsáki, 2007*; *Liu et al., 2019*; *Wikenheiser and Redish, 2013*). For example, animals tend to shift from forward to backward replay from the beginning to the end of a run (*Diba and Buzsáki, 2007*), exhibit more forward replay during sleep (*Wikenheiser and Redish, 2013*), and show biased replay of memories associated with external cues during sleep (*Bendor and Wilson, 2012*). Some of these observations have led investigators to posit distinct processes underlying forward and backward replay (*Koene and Hasselmo, 2008*; *Ambrose et al., 2016*; *Diba and Buzsáki, 2007*; *Foster, 2017*; *Foster and Wilson, 2006*; *Khamassi and Girard, 2020*; *Mattar and Daw, 2018*; *Shin et al., 2019*), with forward replay supporting planning at choice points (*Diba and Buzsáki, 2007*; *Foster, 2017*; *Mattar and Daw, 2018*; *Shin et al., 2019*) and backward replay encoding value expectations from reward outcomes (*Ambrose et al., 2016*; *Foster, 2017*; *Foster and Wilson, 2006*). Here, we evaluate whether CMR-replay can account for these differential patterns under one framework, with replay always reflecting associations between items and contexts.

Animals first acquire experience with a linear track by traversing it to collect a reward. Then, during the pre-run rest recording, forward replay predominates (*Diba and Buzsáki, 2007*). In contrast, backward replay predominates during post-run rest, when the animal consumes its reward (see *Figure 2a*, left; *Diba and Buzsáki, 2007*). We simulated this task by presenting CMR-replay with a sequence of items (Figure 7a), each representing a distinct location. These item representations can be considered to correspond to place cells in rodents, whose activity is typically used to track replay. During post-run rest, we use the final item's encoding context as an external cue for rest replay. For pre-run rest,

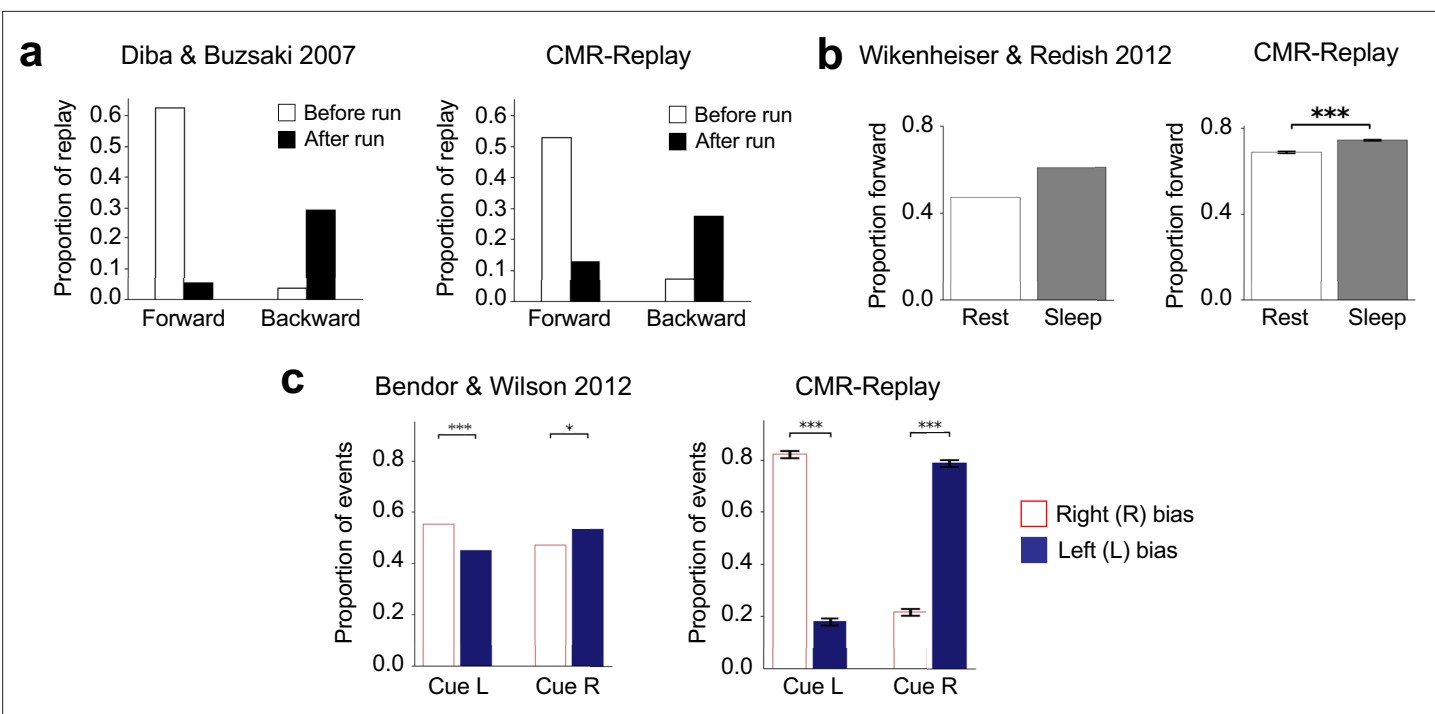

**Figure 2.** Context-dependent variations in memory replay. (**a**) As observed in rodents (left), replay in CMR-replay (right) is predominantly forward at the start of a run and backward at the end of a run on a linear track. (**b**) Consistent with rodent data (left), in CMR-replay (right), the proportion of forward replay is higher during sleep than during awake rest. (**c**) The presence of external cues during sleep biases replay toward their associated memories both in animals (left) and in CMR-replay (right). Error bars represent ±1 standard error of the mean. *p<0.05; **p<0.01; ***p<0.001. Left image in a adapted from *Diba and Buzsáki, 2007*, Nature Publishing Group; left image in **b** adapted from *Wikenheiser and Redish, 2013*, Wiley.

the first item's encoding context serves as the external cue for rest replay. Because of the external cue, awake rest replay initiates disproportionately at the item most strongly associated with the cue (*Figure 1e*), which is consistent with a bias of awake replay to initiate at the resting location (*Davidson et al., 2009*). The conjunction of this initiation bias and the model's contiguity bias entails that replay tends to unroll successively forward from the first item in pre-run rest and backward from the final item in post-run rest (*Figure 2a*, right) as in the data (*Diba and Buzsáki, 2007*). In our model, these patterns are identical to those in our simulation of *Ambrose et al., 2016*, which uses two independent sequences to mimic the two run directions. This is because the drifting context resets before each run sequence is encoded, with the pause between runs acting as an event boundary that prevents the final item of one traversal from associating with the first item of the next, thereby keeping learning in each direction independent. In contrast to the EVB model (*Mattar and Daw, 2018*), CMR-replay captures the graded nature of this phenomenon (*Figure 2a*, right): forward and backward replay appear in both conditions (*Diba and Buzsáki, 2007*). All of the differences between conditions observed here, and across all simulations in the paper, are highly reliable (p<0.001 based on two-tailed t-tests, with runs of the model as random effects factor). Levels of significance are indicated in figures.

As with prior retrieved context models (*Howard and Kahana, 2002*; *Polyn et al., 2009*), CMR-replay encodes stronger forward than backward associations. This asymmetry exists because, during the first encoding of a sequence, an item's associated context contributes only to its ensuing items' encoding contexts. Therefore, after encoding, bringing back an item's associated context is more likely to reactivate its ensuing than preceding items, leading to forward asymmetric replay (Figure 6d, left). Absent external cues, sleep replay is less likely to initiate at the final item than rest replay (*Figure 1e*), allowing for more forward transitions. This leads to more forward replay during sleep than awake rest (*Figure 2b*, right), matching empirical observations (*Foster, 2017*; *Lee and Wilson, 2002*; *Wikenheiser and Redish, 2013*; *Figure 2b*, left). In contrast, the EVB model predicts a predominance of reverse replay before behavior stabilizes (*Mattar and Daw, 2018*). Note that the overall proportion of forward replay is higher in the model than these data, but consistent with that found in *Diba and Buzsáki, 2007*. Our model also predicts that deliberation on a specific memory, such as during planning, could serve to elicit an internal context cue that biases replay: actively recalling the first item of a sequence may favor forward replay, while thinking about the last item may promote backward replay, even when the individual is physically distant from the track. While not explored here, this mechanism presents a potential avenue for future modeling and empirical work.

We next asked whether CMR-replay can simulate targeted memory reactivation (TMR) – the re-presentation of learning-related cues during sleep to encourage reactivation of the associated information. One study employed the TMR paradigm in rodents, associating distinct auditory cues ($L$ and $R$) with left and right traversal of a linear track (*Bendor and Wilson, 2012*). Playing each auditory cue during sleep elicited biased replay of place cell activity in the cued direction. We simulate these findings by encoding two sequences that share a start item. To simulate TMR, we presented a distinct cue item after each sequence's start item during learning (Figure 7e) and re-presented each cue item (through its associated context) as an external cue in sleep. Matching (*Bendor and Wilson, 2012*), CMR-replay preferentially replayed each cue's associated sequence (*Figure 2c*, right). *Bendor and Wilson, 2012*, found that sound cues during sleep did not trigger immediate replay, but instead biased reactivation toward the cued sequence over an extended period of time. While the model does exhibit some replay triggered immediately by the cue, it also captures the sustained bias toward the cued sequence over an extended period.

## Effects of reward

At first glance, our proposal may seem at odds with extensive evidence of the influence of reward on replay (*Ambrose et al., 2016*; *Foster and Wilson, 2006*; *Igata et al., 2021*; *Liu et al., 2019*; *Liu et al., 2021*; *Michon et al., 2019*; *Singer and Frank, 2009*; *Ólafsdóttir et al., 2015*) because CMR-replay neither maintains nor updates value representations during replay. For example, studies suggest that replay over-represents experiences with rewarded or aversive outcomes (*Igata et al., 2021*; *Singer and Frank, 2009*; *Sterpenich et al., 2021*; *Wu et al., 2017*; *Ólafsdóttir et al., 2015*) and awake reverse replay occurs primarily during reward receipt (*Ambrose et al., 2016*; *Diba and Buzsáki, 2007*; *Foster and Wilson, 2006*). Reverse replay's unique sensitivity to reward (*Ambrose et al., 2016*; *Michon et al., 2019*) appears to indicate a functional distinction between forward and

backward replay, with backward replay specialized for learning value-based predictions (*Ambrose et al., 2016*; *Foster and Wilson, 2006*; *Liu et al., 2021*).

We suggest that salience governs encoding rates, which aligns with evidence that salient stimuli bind more strongly to their context (*Anderson, 2005*; *Mackay et al., 2005*; *Mackay et al., 2004*; *Mather, 2007*). Building on models that adopt this assumption (*Cohen and Kahana, 2022*; *Talmi et al., 2019*), CMR-replay updates $M^{fc}$ and $M^{cf}$ at higher rates for salient items, including those with high valence (reward or punishment) or novelty. In CMR-replay, increasing encoding rates strengthens replay in two distinct ways: Enhancing the $M^{cf}$ encoding rate facilitates the reactivation of an item given features of its encoding context as cue, while enhancing the $M^{fc}$ encoding rate facilitates the faithful retrieval of an item's encoding context. Here, we explore whether these mechanisms allow CMR-replay to account for reward-related phenomena.

After visually exploring a T-maze with one arm containing reward, animals preferentially activated sequences representing the rewarded arm during sleep (*Ólafsdóttir et al., 2015*; *Figure 3a*, left). We simulated this task by presenting CMR-replay with two sequences, one with a rewarded final item and the other with a neutral final item (Figure 7d). Due to the influence of encoding rates, replay over-represents the rewarded item compared to the matched neutral item (*Figure 3a*, right) as in empirical observations (*Igata et al., 2021*; *Singer and Frank, 2009*; *Ólafsdóttir et al., 2015*). CMR-replay exhibits this property without the assumption that reward-associated items receive more exposure during encoding (*Diekmann and Cheng, 2023*).

Varying the magnitude of reward at the end of a linear track significantly alters the number of backward but not forward replay events (*Ambrose et al., 2016*; *Figure 3b*, left). Following *Mattar and Daw, 2018*, to disambiguate each location and the direction of a run, we simulated the task with two distinct input sequences, each with a final rewarded item. We manipulated the encoding rate of one rewarded item to be higher (i.e. high reward), lower (i.e. low reward), or identical to that of the reward item in the other sequence (i.e. normal reward). Since the encoding of the rewarded item primarily influences backward replay in post-run rest, we observed differences in the rate of backward but not forward replay between different reward conditions (*Figure 3b*, right), matching empirical observations (*Ambrose et al., 2016*; *Michon et al., 2019*).

CMR-replay's ability to account for the effects of reward supports our proposal that reward modulates the initial encoding of memories to shape subsequent replay. After reward exerts its influence during encoding, prioritized replay of rewarded memories can occur even if reward-related activity is absent. Consistent with our proposal that replay itself does not require value-based computations, sleep's preferential consolidation of reward memories does not seem to require dopaminergic activity

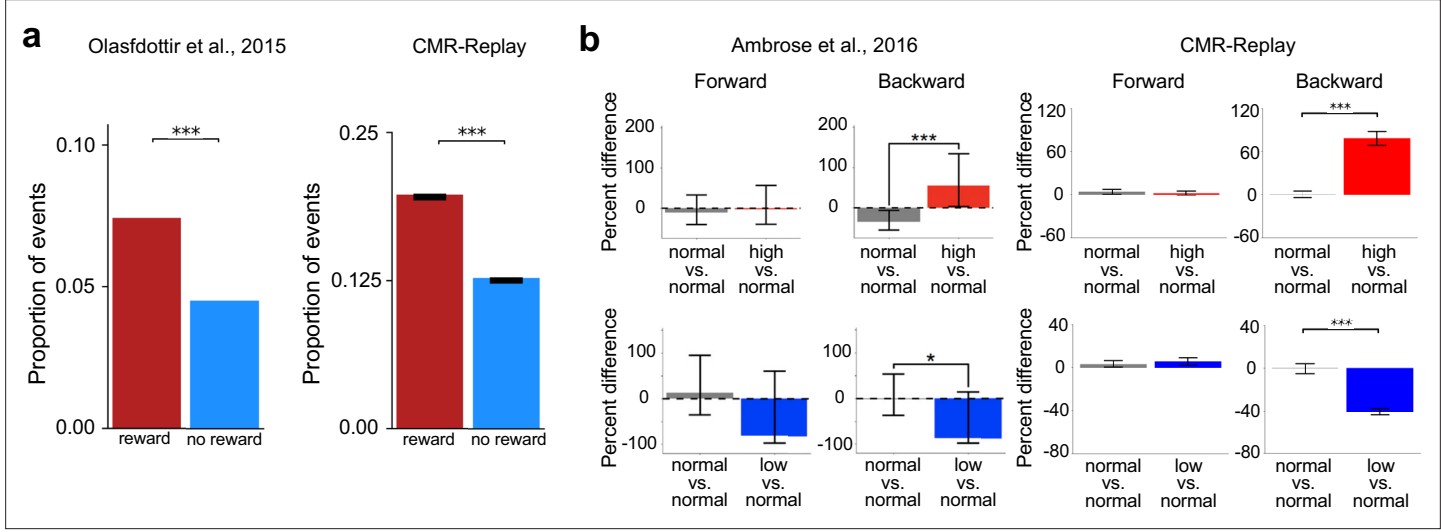

**Figure 3.** Reward leads to over-representation in sleep and modulates the rate of backward replay. (**a**) Sleep over-represents items associated with reward in animals (left) and in CMR-replay (right). Error bars represent ±1 standard error of the mean. (**b**) Varying the magnitude of reward outcome leads to differences in the frequency of backward but not forward replay in animals (left) and CMR-replay (right). In the animal data (left), error bars show 95% confidence intervals. For simulation results (right), error bars show ±1 standard error of the mean. Left image in **a** adapted from *Ólafsdóttir et al., 2015*, eLife; images in the left column adapted from *Ambrose et al., 2016*, Elsevier.

(*Asfestani et al., 2020*), and the coordination between reward responsive neurons and replay-related events is absent in sleep (*Gomperts et al., 2015*). Our model treats reward as simply the salient feature of an item, generating the prediction that non-reward-related salient items should exhibit similar characteristics.

## Replay goes beyond direct recent experience

We next asked whether CMR-replay can account for findings in which animals replay sequences learned outside of their present context. Several studies have established this so-called 'remote replay' phenomenon (*Gupta et al., 2010*; *Karlsson and Frank, 2009*). Here, we describe one such experiment and show how CMR-replay provides an account of its findings. In *Gupta et al., 2010*, animals explored both arms of a T-maze during pre-training. During each subsequent recording session, animals traversed only the left or right arm (L- or R- only conditions) or alternated between them (alternation condition). During reward receipt on the just-explored arm, awake rest exhibited remote replay of the opposite, non-local arm (*Figure 4a*, left: remote replay) across all conditions (*Gupta et al., 2010*). This observation challenges models that prioritize items near the resting location (*Mattar and Daw, 2018*) and recently active neurons (*Buzsáki, 1989*; *Csicsvari et al., 2007*; *Foster and Wilson, 2006*; *O'Neill et al., 2008*) throughout replay. To determine whether CMR-replay can reproduce these results, we presented the model with sequences that overlap for the first few items (representing the central arm of the T-maze; Figure 7c). During each of two simulated 'pre-training' sessions, the model encoded both sequences. We then ran the model through two conditions in an ensuing 'experimental' session, where it encoded either only one (L- or R-only conditions) or both sequences (alternation condition). After encoding the sequences, we simulated

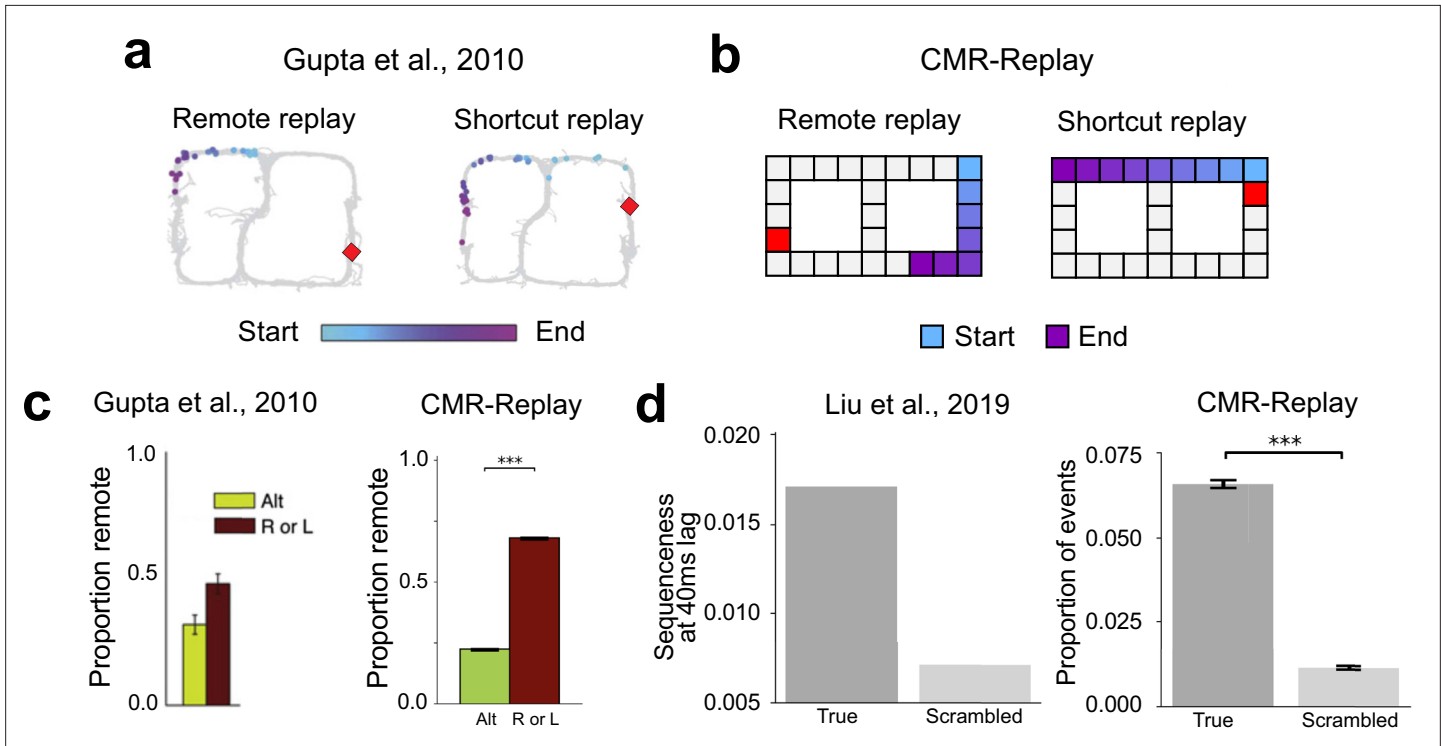

**Figure 4.** Replay activates remote experiences and links temporally separated experiences. (**a**) The two panels show examples of remote and novel (shortcut) replay sequences observed in animals. The colored items indicate the temporal order of the sequences (light blue, early; purple, late). The red item denotes the resting position. (**b**) CMR-replay also generates remote and shortcut rest replay, as illustrated in the replay sequences in the two panels. (**c**) Proportion of replay events that contain remote sequences in animals (left) and in CMR-replay (right). Error bars show ±1 standard error of the mean in the data and model. (**d**) In *Liu et al., 2019*, participants encoded scrambled versions of two true sequences $X_1X_2X_3X_4$ and $Y_1Y_2Y_3Y_4$: $X_1X_2Y_1Y_2$, $X_2X_3Y_2Y_3$, and $X_3X_4Y_3Y_4$ (*Figure 7g*). After learning, human spontaneous neural activity showed stronger evidence of sequential reactivation of the true sequences (left). CMR-replay encoded scrambled sequences as in the experiment. Consistent with empirical observation, subsequent replay in CMR-replay over-represents the true sequences (right). Error bars show ±1 standard error of the mean in the model. Images in **a** adapted from *Gupta et al., 2010*, Elsevier; left image in **c** adapted from *Gupta et al., 2010*, Elsevier; left image in **d** adapted from *Liu et al., 2019*, Elsevier.

reward receipt by presenting CMR-replay with the encoding context of a rewarded item as an external context cue. As in *Gupta et al., 2010*, CMR-replay is able to generate remote replay of the non-local sequence (*Figure 4b*, left; *Figure 4c*, right). When CMR reactivates a non-local item by chance, replay context dramatically shifts by incorporating the non-local item's associated context, thereby triggering a cascade of non-local item reactivation to generate remote replay. Due to its suppression mechanism, CMR-replay is able to capture the higher prevalence of remote replay in L- and R-only conditions (*Figure 4c*, right), which we will unpack in a subsequent section. The occurrence of remote replay, however, does not rely on the suppression mechanism, as the model generates remote replay in the alternation condition where suppression is matched across local and non-local items.

We next examined whether replay in CMR-replay can link temporally separated experiences to form novel sequences that go beyond direct experience (*Barron et al., 2020*; *Diekmann and Cheng, 2023*; *Gupta et al., 2010*; *Kumaran and McClelland, 2012*; *Liu et al., 2019*). *Gupta et al., 2010* showed the occurrence of novel, shortcut replay sequences. These shortcut sequences cut directly across the choice point and link segments of the two arms of a T-maze during rest (*Figure 4a*, right: shortcut replay), even though animals never directly experienced such trajectories (*Gupta et al., 2010*). In our simulation of the study (*Gupta et al., 2010*), CMR-replay also generates novel rest replay that links segments of the two sequences (*Figure 4b*, right): The reactivation of the juncture of the two sequences (the top middle item of Figure 7c) brings back context common to the two sequences, allowing replay to stitch together segments of the two sequences. In line with *Gupta et al., 2010*, shortcut replay appeared at very low rates in CMR-replay (the alternation condition: mean proportion of replay events that contain shortcut sequence = 0.0046; L or R conditions: mean proportion = 0.0062).

*Liu et al., 2019*, showed that replay in humans reorganizes temporally -separated wake inputs. In their first experiment, participants encoded sequences that scrambled pairwise transitions of two true sequences $X_1X_2X_3X_4$ and $Y_1Y_2Y_3Y_4$, $X_1X_2Y_1Y_2$, $X_2X_3Y_2Y_3$, and $X_3X_4Y_3Y_4$. To highlight transitions from the true sequences, the time lag between those transitions (e.g. $X_2X_3$) was shorter than others (e.g. $X_3Y_2$) during presentation. Analyses revealed preferential replay of the true as opposed to the scrambled sequences (*Liu et al., 2019*; *Figure 4d*, left). We simulated the experiment by presenting CMR-replay with sequences of the same structure (Figure 7g), where context drifted more for longer interstimulus intervals. After learning, the model performed replay in the absence of external context cues. The quantification of replay in *Liu et al., 2019*, which reflects statistical evidence of replay decoding based on magnetoencephalography (MEG) data, differs from our measure, where we have direct access to replay without measurement noise. However, qualitatively matching *Liu et al., 2019*, CMR-replay preferentially replays true sequences relative to scrambled sequences (*Figure 4d*, right).

## The influence of experience

Task exposure influences replay, with replay appearing less frequently in familiar as compared with novel environments (*Cheng and Frank, 2008*; *Giri et al., 2019*; *Huelin Gorriz et al., 2023*; *O'Neill et al., 2008*). Task repetition similarly reduces replay (*Diba and Buzsáki, 2007*; *Shin et al., 2019*). After gaining experience along multiple trajectories, animals and humans can exhibit enhanced replay of non-recently explored trajectories (*Carey et al., 2019*; *Gillespie et al., 2021*; *Gupta et al., 2010*; *Wimmer et al., 2023*). Overall, these findings demonstrate a negative relationship between the degree and recency of experience and the frequency of replay. This pattern challenges models in which experience monotonically enhances the reactivation of items (*Buzsáki, 1989*; *Csicsvari et al., 2007*; *Diekmann and Cheng, 2023*; *Foster and Wilson, 2006*; *Mattar and Daw, 2018*; *O'Neill et al., 2008*).

In CMR-replay, experience shapes replay in two opposing ways. First, repetition strengthens $M^{fc}$ and $M^{cf}$, allowing replay to better preserve the temporal structure of waking inputs. Second, by enhancing $M^{fc}$, repetition increases the activity of contexts associated with items. Since CMR-replay suppresses the activity of items at the onset of replay as a function of their activity during learning, repetition increases the downweighting of the activity of task items, reducing their probability of reactivation. Such a suppression mechanism may be adaptive, allowing replay to benefit not only the most recently or strongly encoded items but also to provide opportunities for the consolidation of weaker or older memories, consistent with empirical evidence (*Schapiro et al., 2018*; *Yu et al., 2024*).

The next set of simulations illustrates CMR-replay's account of experience-dependent changes in replay (*Carey et al., 2019*; *Diba and Buzsáki, 2007*; *Gupta et al., 2010*; *Igata et al., 2021*; *Shin et al., 2019*). We first examined how replay changes through repeated encoding of the same inputs following our linear track simulation illustrated in Figure 7a. Here, CMR-replay encodes the same

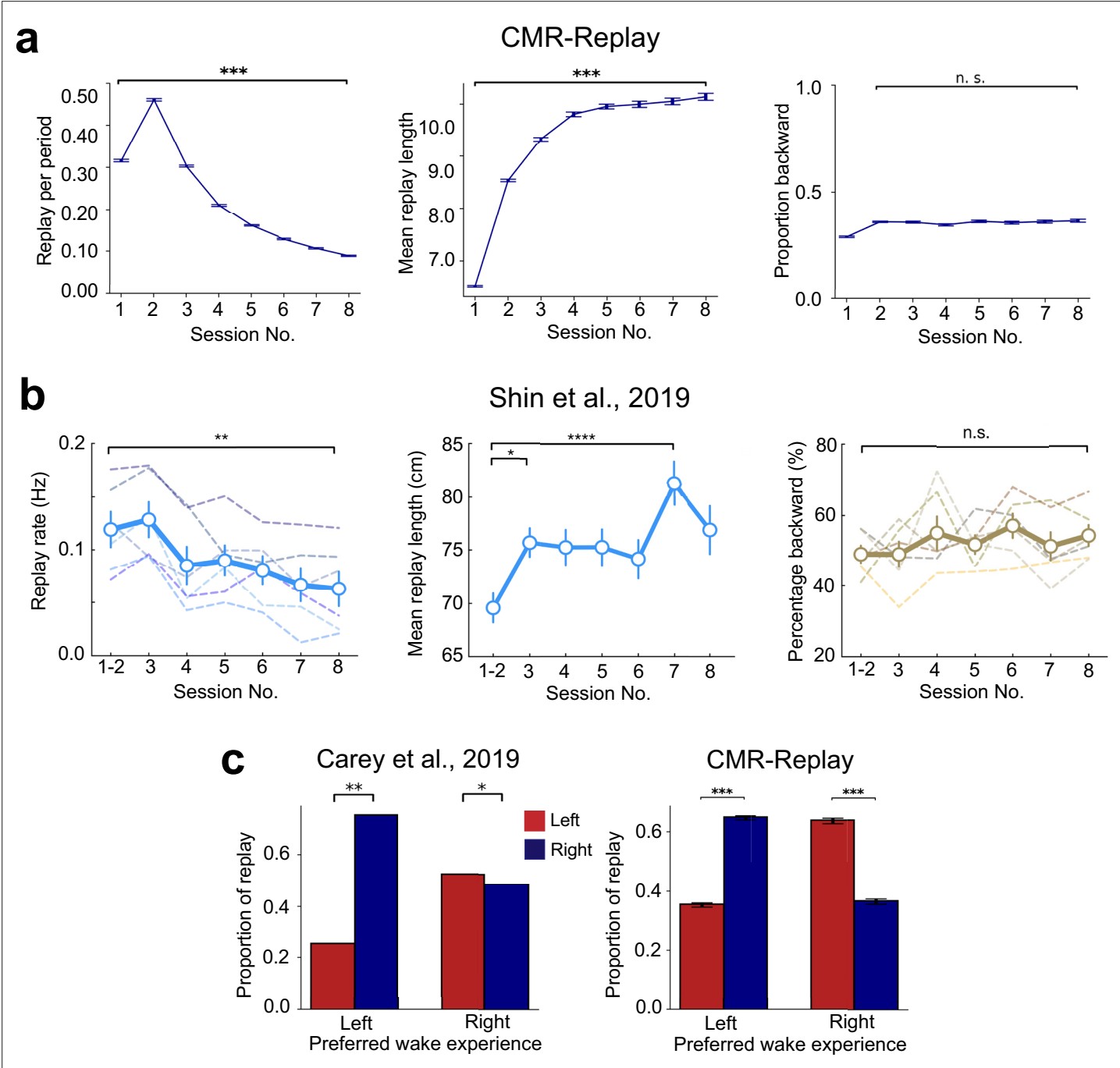

**Figure 5.** Variations in replay as a function of experience. (**a**) In CMR-replay, through repeated exposure to the same task, the frequency of replay events decreases (left), the average length of replay events increases (middle), and the proportion of replay events that are backward remains stable (after a slight initial uptick; right). (**b**) With repeated experience in the same task, animals exhibit lower rates of replay (left) and longer replay sequences (middle), while the proportion of replay events that are backward stays relatively stable (right). (**c**) In a T-maze task, where animals display a preference for traversing a particular arm of the maze, replay more frequently reflects the opposite arm (*Carey et al., 2019*) (left). CMR-replay preferentially replays the right arm after exposure to the left arm and vice versa (right). Error bars show ±1 SEM in all panels. Images in **b** adapted from *Shin et al., 2019*, Elsevier; left image in **c** adapted from *Carey et al., 2019*, Nature Publishing Group.

sequence across learning sessions, with awake rest after each session. Initially, experience increases the prevalence of replay (*Figure 5a*, left). As repetition enhances the suppression of task-related items at the onset of replay, replay frequency subsequently decreases in CMR-replay (*Figure 5a*, left). Through experience, the average length of replay increases (*Figure 5a*, middle), suggesting that repetition strengthens sequence memory in the model. In contrast to the EVB model (*Mattar and Daw, 2018*), which predicts a differential drop in the rate of backward relative to forward replay, the proportion of replay events that are backward does not decrease (*Figure 5a*, right) in CMR-replay. This result highlights that, unlike the EVB model, CMR-replay does not employ distinct variables to drive forward vs. backward replay.

In an experiment where animals learned the same task across eight behavioral sessions, *Shin et al., 2019*, observed a similar pattern of results. As shown in *Figure 5b*, animals exhibited lower rates of replay but longer replay sequences in later sessions (left, middle). As in our CMR-replay simulations, as the rates of forward and backward replay both decrease, the proportion of forward relative to backward replay events remains relatively stable across sessions (right). Furthermore, consistent with reduced reactivation of task-related units in CMR-replay, the study observed decreased reactivation of task-related place cells through experience. In contrast, item reactivation increases monotonically through repetition in other models (*Diekmann and Cheng, 2023*; *Mattar and Daw, 2018*). *Shin et al., 2019*, performed Bayesian decoding to statistically quantify evidence of replay, whereas our analyses directly compare segments of a behavioral sequence with replay sequences. Despite differences between these measures, the patterns of results in the data and in the model match qualitatively. Several other studies using varied experimental procedures have reported similar effects of repeated experience on replay, including a reduction in the prevalence of replay (*Cheng and Frank, 2008*; *Diba and Buzsáki, 2007*), an increase in replay length (*Berners-Lee et al., 2022*), and no reduction in the proportion of replay events that are backward (*Igata et al., 2021*).

In CMR-replay, the activity of retrieved contexts associated with items in a learning session modulates the level of item suppression during ensuing quiescence. As a result, items that get more exposure in a session may receive more suppression than others at the onset of replay, facilitating the reactivation of their competitors. In our simulation of *Gupta et al., 2010*; Figure 7c, in the L- and R-only conditions, since the sequence presented during learning receives more suppression, remote replay is more prevalent than in the alternation condition, where both sequences appear during learning (*Figure 4c*). In the L- or R-only conditions, when CMR-replay performs post-learning replay in the absence of external context cues, replay over-represents the alternative sequence (*Figure 5c*), which aligns with the observation that replay exhibits a bias away from the arm of a T-maze that animals preferred during behavior (*Carey et al., 2019*). This property is also consonant with recent findings that replay preferentially activates non-recent trajectories (*Gillespie et al., 2021*).

## The function of replay

Many have proposed adaptive functions for replay, including for memory consolidation (*Carr et al., 2011*; *Klinzing et al., 2019*; *McClelland et al., 1995*), retrieval (*Carr et al., 2011*; *Jadhav et al., 2012*; *Wimmer et al., 2020*), credit assignment (*Ambrose et al., 2016*; *Foster and Wilson, 2006*; *Momennejad et al., 2018*), and planning (*Mattar and Lengyel, 2022*; *Jensen et al., 2023*; *Pfeiffer and Foster, 2013*). Growing causal evidence suggests that replay benefits memory: TMR enhances memory (*Oudiette and Paller, 2013*), and disrupting SWRs impairs memory (*Ego-Stengel and Wilson, 2010*; *Girardeau et al., 2009*; *Jadhav et al., 2012*). Replay facilitates offline learning in our model by updating $M^{fc}$ and $M^{cf}$ according to the internally reactivated items and contexts during replay. In the following set of simulations, we characterize ways in which replay facilitates memory in the model.

One of the most robust benefits of sleep is on sequence memory, often studied with motor sequence paradigms (*King et al., 2017*). To simulate the impacts of sleep replay on sequence memory, we presented CMR-replay with a five-item sequence and examined whether sleep enhanced memory of the sequence. Before and after sleep, we assessed the proportion of replay sequences that matched the input sequence. The assessment occurred in 'test' periods, where learning rates were set to zero and external cues were absent. In the post-sleep test, CMR-replay generated a higher proportion of sequences matching the correct sequence than in the pre-sleep test (*Figure 6a*), indicating that sleep enhances sequence memory in the model. This motor memory simulation using a model of hippocampal replay is consistent with evidence that hippocampal replay can contribute to

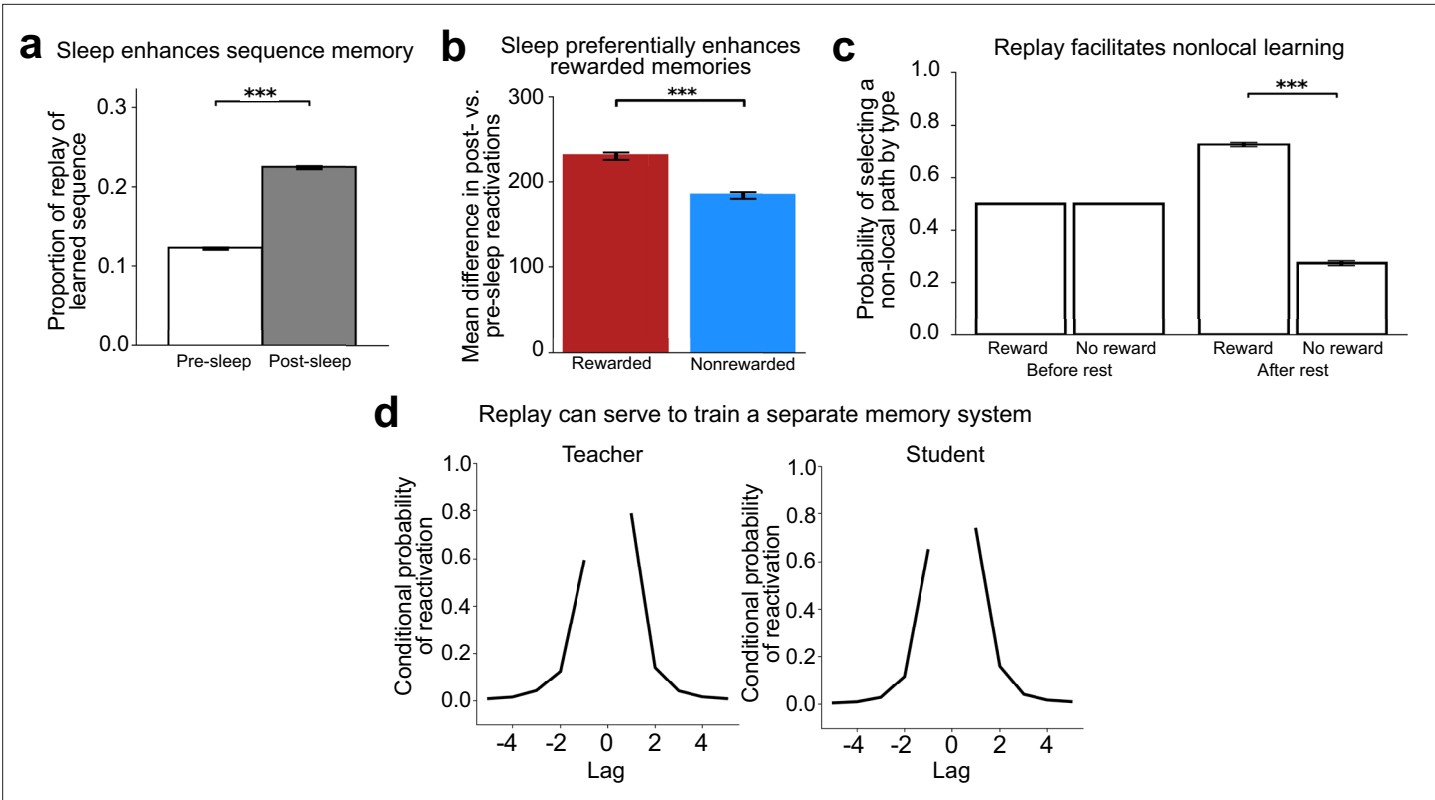

**Figure 6.** Learning from replay. (**a**) Sleep increased the likelihood of reactivating the learned sequence in the correct temporal order in CMR-replay, as seen in an increase in the proportion of replay for learned sequences post-sleep. (**b**) Sleep leads to greater reactivation of rewarded than non-rewarded items, indicating that sleep preferentially strengthens rewarded memories in CMR-replay. (**c**) In the simulation of *Liu et al., 2021*, CMR-replay encoded six sequences, each of which transitioned from one of three start items to one of two end items. After receiving a reward outcome for the end item of a sequence, we simulated a period of rest. After, but not before rest, CMR-replay exhibited a preference for non-local sequences that led to the rewarded item. This preference emerged through rest despite the fact that the model never observed reward in conjunction with those non-local sequences, suggesting that rest replay facilitates non-local learning in the model. (**d**) We trained a 'teacher' CMR-replay model on a sequence of items. After encoding the sequence, the teacher generated replay sequences during sleep. We then trained a separate blank-slate 'student' CMR-replay model exclusively on the teacher's sleep replay sequences. To assess knowledge of the original sequence, we collected sleep replay sequences from both models and assessed the probability that each model reactivates the item at position $i$ + lag of the sequence immediately following the reactivation of the $i$th item of the sequence, conditioned on the availability of the $i$th item for reactivation. Both models demonstrated a tendency to reactivate the item that immediately follows or precedes the just-reactivated item on the original sequence. This result suggests that the student acquired knowledge of the temporal structure of original sequence by encoding only the teacher's replay sequences. Error bars show ±1 SEM.

consolidating memories that are not hippocampally dependent at encoding (*Schapiro et al., 2019*; *Sawangjit et al., 2018*). It is possible that replay in other, more domain-specific areas could also contribute (*Eichenlaub et al., 2020*).

Replay preferentially enhances rewarded memories (*Michon et al., 2019*), and sleep preferentially consolidates salient experiences (*Payne and Kensinger, 2010*; *Payne et al., 2008*). In our simulation of a T-maze with reward in one of the two arms (*Ólafsdóttir et al., 2015*), we also included pre- and post-sleep test periods to assess how sleep in CMR-replay shapes rewarded vs. non-rewarded memory. Through sleep, CMR-replay exhibited a greater increase in its reactivation of the rewarded item compared to a matched neutral item (*Figure 6b*), suggesting that sleep preferentially enhances memory associations for rewarded items in CMR-replay.

A recent study (*Liu et al., 2021*) presented evidence that replay facilitates non-local value learning. Human participants first learned about the structure of six sequences, each of which begins with one of three start items and terminates with one of two end items. Then, for two sequences that share a start state, participants learned that only one of them leads to reward. After a period of rest during which replay was measured with MEG, among the other four sequences (i.e. the non-local sequences), participants exhibited a behavioral preference for sequences that terminate in the end

item associated with reward, despite no direct recent experience with reward in that sequence. The authors suggested that, in accordance with the RL perspective, replay propagates value to associated items, allowing participants to select non-local sequences associated with reward without direct experience. In our simulation of this paradigm, CMR-replay encoded six sequences of the same structure (Figure 7b), with increased encoding rates to simulate reward receipt, as in the simulations above. To simulate awake rest after reward receipt, we presented the encoding context of the rewarded item as an external cue. Before and after rest, we examined the model's preference among the four non-local sequences, by assessing how much the model activated each non-local start item's two ensuing items given the start item's associated context as a cue. After, but not before rest, CMR-replay preferentially activated the item that leads to the rewarded end item (*Figure 6c*). This is because the presence of the rewarded item as an external cue evokes the reactivation of its associated non-local items. In CMR-replay, this preference emerged without value updates, suggesting that replay can facilitate non-local learning by re-organizing memory associations.

There has been much interest in the memory literature in the possibility that hippocampal replay serves to train neocortical systems to represent recent memories (*Alvarez and Squire, 1994*; *Born and Wilhelm, 2012*; *Buzsáki, 1989*; *Carr et al., 2011*; *Hasselmo et al., 1996*; *McClelland et al., 1995*; *Singh et al., 2022*; *Sun et al., 2023*). We explored whether replay in CMR-replay can serve to transfer one model's knowledge to another. After a 'teacher' CMR-replay encodes a sequence, we collected its sleep replay sequences to train a blank-slate 'student' CMR-replay at replay's learning rates. Through this process, the student inherited the contiguity bias of the teacher (*Figure 6d*), suggesting it acquired knowledge of the structure of the teacher's training sequence. This simulation provides a proof of concept that replay in CMR-replay can serve to facilitate memory transfer across systems, in addition to promoting local learning. More broadly, these results resonate with ideas from the Complementary Learning Systems framework (*McClelland et al., 1995*), which proposes that replay allows the neocortex to learn from hippocampal activity by integrating related experiences. We speculate that the offline learning observed in these simulations corresponds to consolidation processes that operate specifically during sleep, when hippocampal-neocortical dynamics are especially tightly coupled (*Klinzing et al., 2019*).

The results outlined above arise from the model's assumption that replay strengthens bidirectional associations between items and contexts to benefit memory. This assumption leads to several predictions about differences across replay types. First, the model predicts that sleep yields different memory benefits compared to rest in the task environment: Sleep is less biased toward initiating replay at specific items, resulting in a more uniform benefit across all memories. Second, the model predicts that forward and backward replay contribute to memory in qualitatively similar ways but tend to benefit different memories. This divergence arises because forward and backward replay exhibit distinct item preferences, with backward replay being more likely to include rewarded items, thereby preferentially benefiting those memories.

## Discussion

What is the nature and function of neural replay? We suggest a simple memory-focused framework that explains a wide array of replay phenomena. First, the brain associates experiences with their encoding contexts at rates that vary according to the salience of each experience. Then, in quiescence, the brain replays by spontaneously reactivating a memory and retrieving its associated context to guide subsequent reactivation. Learning continues to occur during these endogenously generated reactivation events. A model embodying these ideas – CMR-replay – explains many qualitative empirical characteristics of replay and its impacts, including patterns previously interpreted as features of RL computations, and observations that prior models do not explain.

First, CMR-replay demonstrates basic properties of replay that other models exhibit (or could easily accommodate) (*Diekmann and Cheng, 2023*; *Khamassi and Girard, 2020*; *Kumaran and McClelland, 2012*; *Mattar and Daw, 2018*; *McNamee et al., 2021*), including replay's recapitulation of the temporal pattern of past experience during rest and sleep (*Diba and Buzsáki, 2007*; *Wikenheiser and Redish, 2013*), bias toward memories associated with external cues (*Bendor and Wilson, 2012*), and ability to stitch together temporally separated experiences to form novel sequences (*Gupta et al., 2010*; *Liu et al., 2019*). Second, CMR-replay captures findings that have been interpreted as evidence that replay serves value-based RL, including over-representation of memories associated

with reward (*Ólafsdóttir et al., 2015*), reverse replay upon reward receipt (*Diba and Buzsáki, 2007*; *Foster and Wilson, 2006*), and the unique sensitivity of reverse replay to reward magnitude (*Ambrose et al., 2016*). Third, CMR-replay accounts for observations that are not naturally accounted for by prior models, including a stable proportion of backward replay through learning (*Shin et al., 2019*), reduced item reactivation and sequential replay with experience (*Shin et al., 2019*; *Cheng and Frank, 2008*), increased prevalence of forward replay in sleep (*Wikenheiser and Redish, 2013*), enhanced replay outside of the current context (*Gupta et al., 2010*), and a tendency for replay to cover non-behaviorally preferred experiences (*Carey et al., 2019*). Finally, replay facilitates memory in CMR-replay in ways that align with empirical findings (*King et al., 2017*; *Liu et al., 2021*; *Michon et al., 2019*; *Payne et al., 2008*; *Payne and Kensinger, 2010*; *Diamond et al., 2024*). These include the improvement of sequence memory, preferential strengthening of rewarded memories, facilitation of non-local learning, and endogenous training of a separate memory system in the absence of external inputs.

The EVB model and CMR-replay offer different types of explanation for why replay exhibits its disparate characteristics: The EVB model provides an explicitly normative explanation, whereas CMR-replay offers a mechanistic account. The different levels of explanation raise the possibility that CMR-replay could be considered a mechanistic implementation of EVB. Indeed, there are several shared properties between the models (*Diba and Buzsáki, 2007*; *Ólafsdóttir et al., 2015*; *Foster and Wilson, 2006*; *Ambrose et al., 2016*). However, as discussed above, CMR-replay captures observations that appear inconsistent with the EVB model, including the prevalence of non-local replay, the decoupling of replay from behavioral preference, and similar proportions of forward and backward replay over time. In addition to these existing observations, the two models make distinct predictions that can be tested experimentally. For example, in tasks where the goal is positioned in the middle of an arm rather than at its end, CMR-replay predicts a more balanced ratio of forward and reverse replay, whereas the EVB model still predicts a dominance of reverse replay due to backward gain propagation from the reward. This contrast aligns with empirical findings, showing that when the goal is located in the middle of an arm, replay events are more evenly split between forward and reverse directions (*Gupta et al., 2010*), whereas placing the goal at the end of a track produces a stronger bias toward reverse replay (*Diba and Buzsáki, 2007*). Moreover, the EVB model predicts that reward only modulates the rate of replay when it informs potential change in behavior, whereas CMR-replay considers reward as a salient feature that enhances replay by facilitating associations between item and context. Thus, in scenarios where animals cannot choose between competing actions (e.g. when there are only deterministic paths), CMR-replay, but not the EVB model, predicts that reward manipulation would lead to changes in the rate of replay. CMR-replay also predicts that salient non-rewarding events should lead to similar patterns of replay as rewarded events. While not explicitly normative, CMR-replay does offer an explanation of the goal of replay, which is to improve memory through offline local learning and systems interactions. CMR-replay posits that replay may facilitate building a stable, unbiased understanding of the environment useful for many different possible future tasks, some of which may be difficult for an animal to predict and therefore optimize in advance (*Sagiv et al., 2025*).

An ongoing debate concerns to what extent awake replay reflects a process of planning that simulates future scenarios in support of immediate decision-making (*Pfeiffer and Foster, 2013*; *Mattar and Lengyel, 2022*; *Jensen et al., 2023*), vs. to what extent it serves to store, update, and maintain memory without directly guiding behavior (*Dupret et al., 2010*; *Girardeau et al., 2009*; *Joo and Frank, 2018*; *Diba, 2021*). Evidence supporting the planning hypothesis comes from studies that demonstrate enhanced replay of upcoming behavioral trajectories (*Pfeiffer and Foster, 2013*; *Xu et al., 2019*). However, in tasks that track representations of multiple temporally and spatially separated experiences, animals exhibit replay that appears to be decoupled from their behavioral preference (*Carey et al., 2019*; *Gillespie et al., 2021*; *Gupta et al., 2010*). Our model aligns more with the memory perspective, as it is able to capture existing findings without positing that replay serves to optimize behavioral outcome. However, a replay of this kind could at times be read out and used by downstream decision-making systems. For example, recent work argues that the dynamics of the retrieval processes in this class of models could support adaptive choice in sequential decision tasks (*Zhou et al., 2024*). Overall, our framework argues that replay characteristics are primarily driven by memory principles, and that replay serves to strengthen and reorganize memories, which

benefits subsequent – but not necessarily immediate – behavior (*Gillespie et al., 2021*; *Joo and Frank, 2018*).

Many memory consolidation theories are aligned with CMR-replay in suggesting that replay actively strengthens and re-organizes memories (*Antony and Schapiro, 2019*; *Born and Wilhelm, 2012*; *Buzsáki, 1989*; *Cowan et al., 2021*; *Klinzing et al., 2019*; *McClelland et al., 1995*; *Singh et al., 2022*). Contextual binding theory (*Yonelinas et al., 2019*), however, takes a different approach, suggesting that residual encoding-related activity elicits merely epiphenomenal replay as context drifts during quiescence. Our theory echoes this perspective in characterizing replay as an outcome of context-guided processes. However, we diverge from the perspective in suggesting that the emergent replay does significantly benefit memory by strengthening learned associations between items and contexts. Our proposal aligns with a recent TMR study, showing that the recapitulation of items' associated contexts during sleep drives changes in memory in humans (*Schechtman et al., 2023a*). Our model also captures observations of enhanced replay of infrequent and remote experiences, which are in tension with the perspective that replay is primarily guided by recent activity (*Antony and Schapiro, 2019*).

Several recent studies have argued for dominance of semantic associations over temporal associations in the process of human sleep-dependent consolidation (*Schechtman et al., 2023b*; *Liu and Ranganath, 2021*; *Sherman et al., 2025*), with one study observing no role at all for temporal associations (*Schechtman et al., 2023b*). At first glance, these findings appear in tension with our model, where temporal associations drive offline consolidation. Indeed, prior models have accounted for these findings by suppressing temporal context during sleep (*Liu et al., 2024*; *Sherman et al., 2025*). However, earlier models in the CMR lineage have successfully captured the joint contributions of semantic and temporal associations to encoding and retrieval (*Polyn et al., 2009*), and these processes could extend naturally to offline replay. In a paradigm where semantic associations are especially salient during awake learning, the model could weight these associations more and account for greater co-reactivation and sleep-dependent memory benefits for semantically related than temporally related items. Consistent with this idea, *Schechtman et al., 2023b* speculated that their null temporal effects likely reflected the task's emphasis on semantic associations. When temporal associations are more salient and task-relevant, sleep-related benefits for temporally contiguous items are more likely to emerge (*Drosopoulos et al., 2007*; *King et al., 2017*).

Our model has mainly considered replay occurring during SWRs (*Nádasdy et al., 1999*). During active behavior in rodents, ordered place cell sequences also activate during the theta oscillation (theta sequences) (*Johnson and Redish, 2007*). Similar to ripple-based replay, theta sequences manifest in both forward and reverse order (*Wang et al., 2020*), initiate at the animal's location, extend further into upcoming locations through experience (*Blum and Abbott, 1996*; *Jensen and Lisman, 2005*; *Mehta et al., 1997*; *Redish and Touretzky, 1998*), cluster around behaviorally relevant items (*Wikenheiser and Redish, 2015*), and have been proposed to correspond to cued memory retrieval (*Lisman and Redish, 2009*). These parallels lead us to speculate that the context-driven mechanisms we have laid out for findings of replay mainly during SWRs may also be relevant in understanding theta sequences, though future work will be needed to extend the model into this domain.

Our model implements an activity-dependent suppression mechanism that, at the onset of each offline replay event, assigns each item a selection probability inversely proportional to its activation during preceding wakefulness. The brain could implement this by tagging each memory trace in proportion to its recent activation; during consolidation, that tag would then regulate starting replay probability, making highly active items less likely to be reactivated. A recent paper found that replay avoids recently traversed trajectories through awake spike-frequency adaptation (*Mallory et al., 2025*), which could implement this kind of mechanism. In our simulations, this suppression is essential for capturing the inverse relationship between replay frequency and prior experience. Note that, unlike the synaptic homeostasis hypothesis (*Tononi and Cirelli, 2006*), which proposes that the brain globally downscales synaptic weights during sleep, this mechanism leaves synaptic weights unchanged and instead biases the selection process during replay.

Another important area for future work is to investigate how components of CMR-replay map onto map onto brain areas and their interactions. First, our model employs a series of bidirectional operations between context and item representations to generate replay. These operations might be implemented within the recurrent connections of CA3 in the case of temporally compressed SWR replay.

It is possible that these interactions could also play out across the 'big loop' of the hippocampus (*Kumaran and McClelland, 2012*) or within cortical circuits (*Euston et al., 2007*; *O'Neill et al., 2017*; *Vaz et al., 2020*; *Wittkuhn and Schuck, 2021*), which could correspond to slower forms of replay (*Liu et al., 2021*; *Denovellis et al., 2021*). Second, in CMR-replay, the key distinction between awake rest and sleep is whether external inputs bias the initial activation state $a_0$ of the replay process. This simple distinction allows the model to account for key empirical differences between awake and sleep replay. The distinction aligns with the observation that disrupting entorhinal cortex input to the hippocampus affects only awake replay, whereas manipulating hippocampal subfield CA3 activity affects both awake and sleep replay (*Yamamoto and Tonegawa, 2017*). Our view aligns with the theory proposed by *Hasselmo, 1999*, which suggests that the degree of hippocampal activity driven by external inputs differs between waking and sleep states: High acetylcholine levels during wakefulness bias activity into the hippocampus, while low acetylcholine levels during slow-wave sleep allow hippocampal activity to influence other brain regions. Other factors, such as task engagement, may also modulate the influence of external inputs on replay (*Ólafsdóttir et al., 2017*). Third, in quiescence, we posit that the hippocampus can serve as a 'teacher' that endogenously samples memory sequences to help establish these associations in neocortical areas, with local context-item loops within the teacher and student areas. This process may be most likely to take place during NREM sleep, when ripples, spindles, and slow oscillations may coordinate replay between the hippocampus and neocortical areas (*Klinzing et al., 2019*).

Our current simulations have focused on NREM, since the vast majority of electrophysiological studies of sleep replay have identified replay events in this stage. We have proposed in other work that replay during REM sleep may provide a complementary role to NREM sleep, allowing neocortical areas to reinstate remote, already-consolidated memories that need to be integrated with the memories that were recently encoded in the hippocampus and replayed during NREM (*Singh et al., 2022*). An extension of our model could undertake this kind of continual learning setup, where the student, but not teacher, network retains remote memories, and the driver of replay alternates between hippocampus (NREM) and cortex (REM) over the course of a night of simulated sleep. Other differences between stages of sleep and between sleep and wake states are likely to become important for a full account of how replay impacts memory. Our current model parsimoniously explains a range of differences between awake and sleep replay by assuming simple differences in initial conditions, but we expect many more characteristics of these states (e.g. neural activity levels, oscillatory profiles, neurotransmitter levels, etc.) will be useful to incorporate in the future.

There exists a range of computational models of replay that vary in biological plausibility, from biologically detailed frameworks capturing synaptic and spiking dynamics to more abstract, algorithmic approaches (*Ecker et al., 2022*; *Koene and Hasselmo, 2008*; *Hasselmo, 2008*; *Mattar and Daw, 2018*; *Levenstein et al., 2024*; *Shen and McNaughton, 1996*; *Spens and Burgess, 2024*; *Diekmann and Cheng, 2023*; *Haga and Fukai, 2018*; *Khamassi and Girard, 2020*; *Kumaran and McClelland, 2012*; *Jensen et al., 2023*; *Barry and Love, 2022*; *Káli and Dayan, 2004*; *McClelland et al., 1995*; *Santoro et al., 2016*; *Singh et al., 2022*; *Sagiv et al., 2025*). CMR-replay falls toward the latter end of the spectrum, providing a high-level description of a mechanism that accounts for replay phenomena without simulating realistic spiking, synaptic, or membrane potential mechanisms. Aspects of the model, such as its lack of regulation of the cumulative positive weight changes that can accrue through repeated replay, are biologically implausible (as biological learning results in both increases and decreases in synaptic weights) and limit the ability to engage with certain forms of low-level neural data (e.g. changes in spine density over sleep periods; *de Vivo et al., 2017*; *Maret et al., 2011*). It will be useful for future work to explore model variants with more elements of biological plausibility to help span across these styles of model and levels of analysis.

Our theory builds on a lineage of memory-focused models, demonstrating the power of this perspective in explaining phenomena that have often been attributed to the optimization of value-based predictions. In this work, we focus on CMR-replay, which exemplifies the memory-centric approach through a set of simple and interpretable mechanisms that we believe are broadly applicable across memory domains. Elements of CMR-replay share similarities with other models that adopt a memory-focused perspective. The model learns distributed context representations whose overlaps encode associations among items, echoing associative learning theories in which overlapping patterns capture stimulus similarity and learned associations (*Mclaren and*

*Mackintosh, 2002*). Context evolves through bidirectional interactions between items and their contextual representations, mirroring the dynamics found in recurrent neural networks (*Haga and Fukai, 2018*; *Levenstein et al., 2024*). These related approaches have not been shown to account for the present set of replay findings and lack mechanisms – such as reward-modulated encoding and experience-dependent suppression – that our simulations suggest are essential for capturing these phenomena, but we believe these mechanisms could be integrated into architectures like recurrent neural networks (*Levenstein et al., 2024*) in the future to support a broader range of replay dynamics. Furthermore, by building on established models of memory retrieval, CMR-replay naturally aligns with recent theories, suggesting that offline reactivation and online retrieval may have similar underlying mechanisms and utility for behavior (*Antony et al., 2017*). In sum, our theory unifies a wide range of empirical findings under a memory-focused account, offering an integrative and mechanistic framework for how the brain initially encodes and later replays memories to support behavior.

## Methods

### Representation and initialization

CMR-replay comprises four components as in previous retrieved-context models (*Howard and Kahana, 2002*; *Polyn et al., 2009*): item ($f$), context ($c$), item-to-context associations ($M^{fc}$), and context-to-item associations ($M^{cf}$). During both awake encoding and replay, $f$ represents the current item (i.e. an external input presented during awake learning or a reactivated item during replay). $c$ represents a recency-weighted sum of contexts associated with past and present items. During both awake encoding and replay, CMR-replay associates each item with its current encoding context by updating two sets of weights $M^{fc}$ and $M^{cf}$. $M^{fc}$ represents item-to-context associations that support the retrieval of an item's associated context. $M^{cf}$ represents context-to-item associations that enable the retrieval of a context's associated items.

CMR-replay employs a one-hot representation of $f$ (i.e. a localist item representation): Each item is represented by a vector of length $n$ in which only the unit representing the item is on (i.e. has an activity of 1) and all other units are off. As illustrated in *Figure 1*, in addition to task-related items shown as inputs during learning, we include task-irrelevant items that do not appear as inputs during awake encoding, but compete with task-relevant items for reactivation during replay, thereby introducing competition during retrieval and reactivation and mimicking memories of items that do not belong to an ongoing experiment. We use $n_{task}$, $n_{non-task}$, and $n$ to, respectively, denote the number of task-related items, the number of task-irrelevant items, and the total number of items (i.e. the sum of $n_{task}$ and $n_{non-task}$). To allow for sufficient competition between task-related and task-irrelevant items, we set $n_{non-task}$ to be roughly one half of $n_{task}$ (i.e. rounded up when $n_{task}$ is odd). We note that the particular ratio of $n_{non-task}$ to $n_{task}$ is not critical to the pattern of results in our simulations.

In each simulation, $M^{fc}$ and $M^{cf}$ are initialized as identity matrices of rank $n$, which are scaled, respectively, by 1.0 and 0.7. These scaling factors were chosen to qualitatively match the empirically observed proportions of forward and backward replay in different conditions (though the forward/backward asymmetry is always observed in the model). Our initialization of these two matrices as identity matrices differs from the initialization strategy in prior work (*Cohen and Kahana, 2022*; *Howard and Kahana, 2002*; *Lohnas et al., 2015*; *Polyn et al., 2009*), where $M^{fc}$ and $M^{cf}$ are initialized to reflect pre-experimental similarity among items. Given such initializations, prior to learning, $M^{fc}$ maps distinct items onto orthogonal context features, and $M^{cf}$ maps each context feature to a different item. Before the model encodes each input sequence, $c$ is reset as a zero vector of length $n$. Resetting contexts in between sequence presentations demarcates boundaries between discrete events as in prior work (*Pu et al., 2022*).

### Awake encoding
#### Context drift

During awake encoding, the model encodes a sequence of distinct items by associating them with a drifting context. At each timestep $t$, CMR-replay retrieves the current item $f_t$'s associated context $c_{f_t}$ via item-to-context matrix $M^{fc}_{t-1}$ (i.e. the $M^{fc}$ matrix after the previous timestep) according to:

$$c_{f_t} = \frac{M_{t-1}^{fc} f_t}{\| M_{t-1}^{fc} f_t \|} \tag{1}$$

Given $c_{f_t}$, the context layer incorporates $c_{f_t}$ and downweights previous items' associated contexts according to:

$$c_t = \rho c_{t-1} + \beta c_{f_t} \tag{2}$$

where $c_{t-1}$ represents the context before $f_t$ is presented. $\rho$ and $\beta$ determine the relative contribution of $c_{t-1}$ and $c_{f_t}$ to $c_t$. To ensure that $c_t$ has a unit length, $\rho$ is computed according to:

$$\rho = \sqrt{1 + \beta^2[(c_{t-1}c_{f_t})^2 - 1]} - \beta(c_{t-1}c_{f_t}) \tag{3}$$

Therefore, the context layer is a drifting, recency-weighted average of contexts associated with items presented up to timestep $t$. Operations that drive context drift in our simulations, including those specified by *Equations 1–3*, are identical to those in prior work (*Cohen and Kahana, 2022*; *Polyn et al., 2009*; *Lohnas et al., 2015*). In all simulations, $\beta$ is 0.75 (similar to drift rates for temporal context features reported in *Polyn et al., 2009*), except when distractors cause context drift in the simulation of *Liu et al., 2019*.

## Updating $M^{fc}$ and $M^{cf}$

Each time the context drifts, CMR-replay updates $M^{fc}$ and $M^{cf}$ to strengthen associations between the current item ($f_t$) and context ($c_t$). The model updates $M^{fc}$ and $M^{cf}$ using a standard Hebbian learning rule according to:

$$\triangle M^{fc} = \gamma_{fc} c_t f_t^T \tag{4}$$

$$\triangle M^{cf} = \gamma_{cf} f_t c_t^T \tag{5}$$

in which $\gamma_{fc}$ and $\gamma_{cf}$ control the rates at which $M^{fc}$ and $M^{cf}$ are updated.

## Varying awake encoding rates according to salience

CMR-replay assumes that salience modulates the magnitude of $\gamma_{fc}$ and $\gamma_{cf}$. Building on prior work (*Cohen and Kahana, 2022*; *Talmi et al., 2019*), the model assumes that $\gamma_{fc}$ and $\gamma_{cf}$ for awake encoding are higher for salient items, including those that are rewarding (i.e. directly associated with a reward) or novel (i.e. received less exposure), than for other items. Salience modulates $\gamma_{fc}$ and $\gamma_{cf}$ only for the current item and context: It does not modify $\gamma_{fc}$ and $\gamma_{cf}$ for items that precede the salient item. Higher encoding rates allow salient items to associate with their encoding contexts more quickly than other items. This shapes subsequent replay in two distinct ways. First, with a higher $M^{cf}$ encoding rate for a salient item, the model will more strongly activate the item given its encoding context as a cue after awake encoding. Second, with a higher $M^{fc}$ encoding rate, the model will be better able to faithfully retrieve the item's encoding context after awake encoding. For all simulations, the base learning rates $\gamma_{fc}$ and $\gamma_{cf}$ are 1.0. For rewarded items, learning rates vary according to the magnitude of reward: Learning rates $\gamma_{fc_{\text{low}}}$ and $\gamma_{cf_{\text{low}}}$ are 1.0 for items associated with low reward, $\gamma_{fc_{\text{normal}}}$ and $\gamma_{cf_{\text{normal}}}$ are 1.5 for those with standard reward, and $\gamma_{fc_{\text{high}}}$ and $\gamma_{cf_{\text{high}}}$ are 2.0 for those with high reward. These values are chosen to align with prior work (*Cohen and Kahana, 2022*; *Talmi et al., 2019*), in which these scaling factors are 1.0 for items that evoke no arousal (*Cohen and Kahana, 2022*) and greater than 1.0 for those assumed to evoke emotional arousal (*Cohen and Kahana, 2022*; *Talmi et al., 2019*). When an input becomes less novel as it is repeated across sessions, its learning rate in the present session is $\frac{\gamma}{i}$, where $i$ is the index of the current session and $\gamma$ is its initial learning rate. The reduction in learning rates with repetition is important for maintaining a degree of stochasticity in the model's replay during task repetition, since linearly increasing weights would, through the softmax choice rule, exponentially amplify differences in item reactivation probabilities, sharply reducing variability in replay.

## Replay

After each session of awake encoding, CMR-replay autonomously reactivates items during a number of replay periods. For simplicity, we assumed that the number of replay periods is fixed, rather than determined by task-related variables.

### Initial item reactivation

At the onset (i.e. $t = 0$) of each replay period, CMR-replay selects an item from a probability distribution $a_0$, which represents spontaneous activities across items during awake rest or sleep (*Figure 1e*). To simulate the relative lack of external sensory input in sleep, we fill $a_0$ with random activities across all items. By contrast, for awake rest simulations, we make available an external context cue $c_{external}$ (e.g. the task context of the animal's current location) representing where the model is 'resting', which evokes additional activities that bias $a_0$ toward the item most strongly associated with the context cue. Concretely, in $a_0$, the activity of $i$th unit is:

$$[a_0]_i = \frac{[a_0^{random} + \lambda a_0^{evoked}]_i}{\sum_{j=1}^n [a_0^{random} + \lambda a_0^{evoked}]_j} \tag{6}$$

where $a_0^{random}$ represents internal activity that we simulated as random noise, $a_0^{evoked}$ is activity that $c_{external}$ evokes according to *Equation 7*, and $n$ is the total number of items in a simulation. $a_0^{random}$ is a vector of size $n$ whose elements are independently and uniformly drawn from the interval [0, 0.001]. CMR-replay samples an item $f_0$ from $a_0$ for the initial item reactivation.

### Subsequent reactivations

Given $f_0$, the model reinstates its associated context as $c_0$ according to *Equation 1* (*Figure 1f*). This allows the model to engage in a series of autonomous reactivations (*Figure 1a*).

At each timestep $t \geq 1$, CMR-replay reactivates another item without replacement (i.e. by excluding items reactivated at previous timesteps). In particular, the model uses the previous context $c_{t-1}$ as a context cue to evoke a probability distribution $a_t$ that excludes items reactivated prior to $t$. Let $U_t$ denote the set of items that have not yet been reactivated. The probability of each item in $U_t$ is:

$$[a_t^{evoked}]_i = \frac{\exp([M^{cf} c_{cue}]_i / T_t)}{\sum_{\forall f_j \in U_t} \exp([M^{cf} c_{cue}]_j / T_t)} \tag{7}$$

where $T_t$ is a temperature parameter that scales the relative difference of activities in $a_t^{evoked}$ is 0.1 and $T_t$ is 0.14 for all $t \geq 1$. For $t \geq 1$, $c_{cue} = c_{t-1}$ and $a_t = a_t^{evoked}$. In contrast, at $t = 0$, $c_{cue} = c_{external}$ and $a_0$ is a combination of $a_0^{random}$ and $a_0^{evoked}$. Based on *Equation 7*, all $U_t$ items have nonzero activity in $a_t$. From $a_t$, the model samples an item $f_t$.

Given $f_t$, the model performs three operations that parallel the operations performed during awake encoding. First, it reinstates $f_t$'s associated context $c_{f_t}$ via $M^{fc}$ according to *Equation 1*. Then, $c_{f_t}$ induces a drift in context according to *Equation 2*, forming a new context $c_t$, which will guide the reactivation at the next timestep. Finally, the model strengthens the association between $f_t$ and the current context $c_t$ by updating $M^{fc}$ and $M^{cf}$ according to *Equation 4* and *Equation 5*. Compared to awake encoding, the model performs these updates at a slower learning rate $\gamma_{replay}$ of 0.001 during replay. This slower learning rate allows replay to preserve and strengthen memories despite the noisy nature of replay sequences.

At each timestep, the replay period ends either with a stop probability of 0.1, an arbitrary choice made to constrain replay episode length probabilistically, or if a task-irrelevant item becomes reactivated. Future work could explore the implications of varying this parameter more systematically. Following prior work (*Mattar and Daw, 2018*), we consider replayed sequences (one per replay period) with consecutive segments of length five or greater that preserve the contiguity of wake inputs as replay events.

### An experience-dependent suppression mechanism

CMR-replay employs a mechanism that suppresses the activity of items in $a_0$ according to the magnitude of context activity in the preceding awake encoding period. This mechanism differs from related

mechanisms in prior work (*Lohnas et al., 2015*; *Polyn et al., 2009*), which scale the degree of competition among items during recall. This experience-dependent suppression mechanism is distinct from the reduction of learning rates through repetition; it does not modulate the update of memory associations but exclusively governs which items are most likely to initiate replay. For each item $f$ presented in a wake learning session, its activity in $a_0$ is multiplied by:

$$\omega = exp(-\|c_f\|) \tag{8}$$

where $\|c_f\|$ is the Euclidean norm of the item's retrieved context vector $c_f$ in the recent wake learning session given by:

$$\|c_f\| = \sqrt{\sum_{i=1}^{m} c_{f_i}^2} \tag{9}$$

where $m$ is the number of units in $c_f$. For items not shown in the wake session, $C$ is 0.0 and thus $\omega$ is 1.0.

## Task simulations

During awake learning, CMR-replay encodes sequences of items, representing spatial trajectories or other stimulus sequences (*Figure 7*). After awake encoding, the model participates in a number of awake rest or sleep replay periods. In each simulation, for each condition, we ran 100 instantiations of the model with the same initialization. Across these replay periods, each of which produces one reactivated sequence, we compute the proportion of sequences that qualify as significant replay events. As described above, a significant replay event is defined as a reactivated sequence that includes a consecutive segment of five or more items that preserves the contiguity of the wake inputs. For simulations assessing forward vs. backward replay sequences, we further report the proportion of significant replay events occurring in the forward direction (i.e. in the same order as the wake inputs) and the backward direction, where the order is reversed. Variability in replay sequences across models arises from the stochastic nature of the replay process. Due to the variability in replay sequences, different models develop distinct $M^{fc}$ and $M^{cf}$ as they learn from replay. Unlike prior work that identified the best-fitting parameters for each simulation (*Polyn et al., 2009*; *Lohnas et al., 2015*; *Cohen and Kahana, 2022*), CMR-replay employs the same set of model parameters across simulations with varying input structures.

In the simulation that examines context-dependent forward and backward replay through experience (*Figures 2a and 5a*), CMR-replay encodes an input sequence shown in *Figure 7a*, which simulates a linear track run with no ambiguity in the direction of inputs, over eight awake learning sessions (as in *Shin et al., 2019*). In this simulation, learning rates for the rewarded item are $\gamma_{fc_{normal}}$ and $\gamma_{cf_{normal}}$. After each wake learning session, we simulate 500 awake rest replay periods at the end of a run followed by another 500 periods at the start of a run. For rest at the end of a run, $c_{external}$ is the context associated with the final item in the sequence. For rest at the start of a run, $c_{external}$ is the context associated with the start item. In this simulation, the learning rates progressively decrease across sessions, as described above. The proportion of replay shown in *Figure 2a* represents the proportion of reactivated sequences categorized as significant forward vs. backward replay sequences. *Figure 5a* depicts the mean number of significant replay sequences per replay period.

In the simulation that contrasts forward and backward replay between rest and sleep (*Figure 2b*), the model encodes the input sequence shown in *Figure 7a* for a single session. After encoding, each model either participates in 1000 awake rest or sleep replay periods, with 100 models in each condition (i.e. awake rest or sleep). *Figure 2b* reports the proportion of significant replay sequences categorized as forward replay.

In the simulation of TMR (*Figure 2c*), each model encodes two sequences shown in *Figure 7e* in a randomized order in a session of wake learning. During input presentation, for each sequence, a separate cue item (cue L or cue R) is presented immediately after the start item. The models encode the goal item at rates $\gamma_{fc_{normal}}$ and $\gamma_{cf_{normal}}$. After wake learning, each model engages in 500 sleep replay periods. In each replay period, the context associated with cue L or cue R is randomly presented as $c_{external}$. *Figure 2c* presents the proportion of significant replay sequences corresponding to the sequence associated with cue L vs. that associated with cue R.

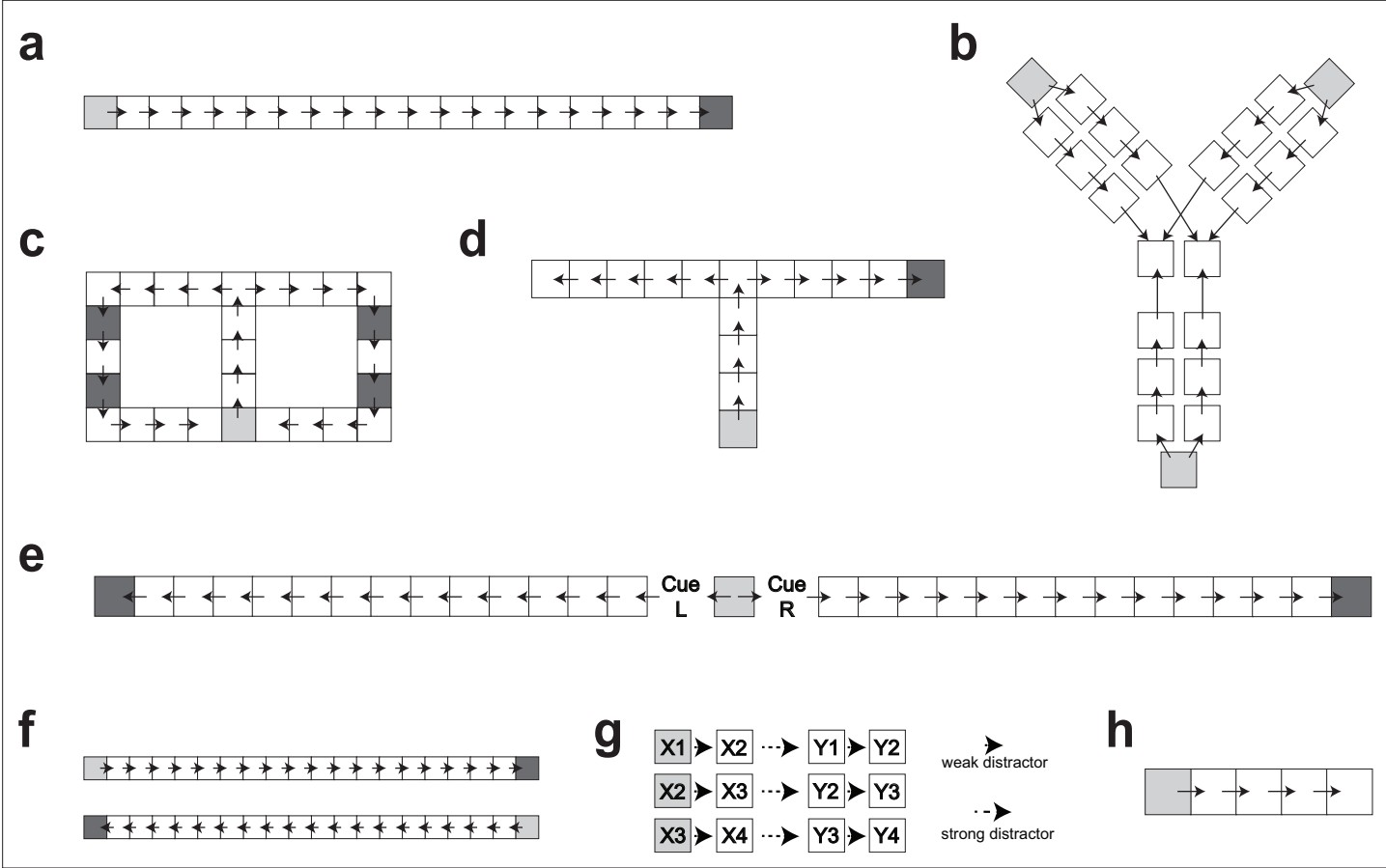

**Figure 7.** Task simulations. Each enclosed box corresponds to a unique item. Each arrow represents a valid transition between two items. Each dashed arrow represents a distractor that causes a drift in context between two items. Task sequences initiate at light gray boxes. Dark gray boxes represent salient items in each task. For tasks with multiple valid sequences, the order in which sequences are presented is randomized. (**a**) Simulation of a linear track. (**b**) Simulation of the task in **Liu et al., 2021**. (**c**) Simulation of a two-choice T-maze. (**d**) Simulation of a T-maze. (**e**) Simulation of the task in **Bendor and Wilson, 2012**. (**f**) Simulation of a linear track task with distinct directions of travel. (**g**) Simulation of input sequences in **Liu et al., 2019**. (**h**) Simulation of a fixed-item sequence.

In the simulation that contrasts replay of rewarded vs. non-rewarded items (**Figures 3a and 6b**), each model encodes two sequences shown in **Figure 7d** in a randomized order in a session of wake learning. The models encode the goal item at rates $\gamma_{fc_{normal}}$ and $\gamma_{cf_{normal}}$. After wake learning, each model engages in an extended phase with 5000 sleep replay periods. To quantify changes in memory through sleep, in each model, we additionally simulated 5000 replay periods before and after extended sleep with no learning (i.e. $M^{fc}$ and $M^{cf}$ are not updated) and no $c_{external}$. **Figure 3a** presents the proportion of replay periods reactivating the rewarded goal vs. the non-rewarded goal.

In the simulation of forward and backward replay with different levels of reward (**Figure 3b**), the model encodes two sequences (**Figure 7f**) in a randomized order in a single session. The inclusion of two disjoint sequences follows the approach in **Mattar and Daw, 2018**, which simulates different directions of travel to distinguish place cells with directional preference for replay decoding in animal studies. The simulation consists of three conditions: normal vs. normal reward, low vs. normal reward, and high vs. normal reward. In the normal vs. normal condition, each model encodes goal locations in both sequences at rates $\gamma_{fc_{normal}}$ and $\gamma_{cf_{normal}}$. In the low vs. normal condition, each model encodes the goal location at rates $\gamma_{fc_{low}}$ and $\gamma_{cf_{low}}$ for one sequence and at rates $\gamma_{fc_{normal}}$ and $\gamma_{cf_{normal}}$ for the other. Finally, in the high vs. normal condition, each model encodes the goal location at rates $\gamma_{fc_{high}}$ and $\gamma_{cf_{high}}$ for one sequence and at rates $\gamma_{fc_{normal}}$ and $\gamma_{cf_{normal}}$ for the other. After encoding a sequence, we simulate 500 awake rest replay periods at the end of a run followed by another 500 at the start of a run. **Figure 3b** presents relative differences in the proportion of significant replay sequences that are categorized as forward vs. backward.

In the simulation of remote replay, shortcut replay, and the over-representation of non-behaviorally preferred experiences in replay (*Figures 4b, c and 5c*), each model encodes two sequences (*Figure 7c*) in a randomized order in a total of three sessions. Learning rates for each goal location are $\gamma_{fc_{\text{normal}}}$ and $\gamma_{cf_{\text{normal}}}$. In these simulations, we treat the first two sessions as the period in which an animal is pre-trained extensively on the task. After wake learning in the third session, for the results shown in *Figure 4b*, each model engages in 500 awake rest replay periods at each of the four goal locations in a randomized order. For the results shown in *Figure 5c*, to simulate replay away from the task environment, each model engages in 500 replay periods with no external context cue. *Figure 4c* presents the proportion of significant replay sequences that are categorized as remote replay.

In the simulation of *Liu et al., 2019*; *Figure 4d*, each model encodes three sequences (*Figure 7g*) shown in a randomized order. These three sequences $X_1X_2Y_1Y_2$, $X_2X_3Y_2Y_3$, and $X_3X_4Y_3Y_4$ are scrambled versions of pairwise transitions from true sequences $X_1X_2X_3X_4$ and $Y_1Y_2Y_3Y_4$. A distractor item, which is a distinct item that does not participate in replay, induces context drift between successive items. The item induces context drift at a $\beta$ of 0.3 for transitions that exist in the true sequences and at a $\beta$ of 0.99 for transitions that do not exist in the true sequences (simulating the longer interstimulus interval between these transitions in the experiment). *Figure 4d* shows the proportion of replay sequences categorized as significant replay sequences matching the true vs. scrambled sequences.

In the simulation that examines sequence memory through sleep (*Figure 6a*), each model encodes a five-item sequence (*Figure 7h*) in a session. After wake learning, each model participates in an extended period of sleep with 5000 replay periods. In each model, we additionally simulated 5000 replay periods before and after extended sleep with no learning (i.e. $M^{fc}$ and $M^{cf}$ are not updated) and no $c_{external}$. *Figure 6a* shows the proportion of replay sequences categorized as significant replay sequences matching the five-item sequence both prior to and following sleep.

In the simulation that examines replay's role in non-local learning (*Figure 6c*), each model encodes six sequences (*Figure 7b*) in a randomized order in a session. Sequences in this simulation consist of three start states and two end states. Each start state has a unique sequence that connects it to each of the two end states. The model encodes the final item in the final sequence at rates $\gamma_{fc_{\text{high}}}$ and $\gamma_{cf_{\text{high}}}$ and encodes all other items at base learning rates $\gamma_{fc}$ and $\gamma_{cf}$. After the encoding of all six sequences, each model participates in 5000 awake rest replay periods with the final item's associated context as $c_{external}$. Before and after the rest period, we evaluated the model's preference for non-local sequence trajectories associated with rewarded items. Specifically, we assessed the model's activation of each non-local start item's two ensuing items given the start item's associated context as a cue. These activations were then normalized into a probability distribution using a softmax transformation, providing a measure of the model's choice preference between the reward-associated path and the non-rewarded path. *Figure 6c* illustrates how these probabilities change through rest.

In the simulation of teacher and student CMR-replay (*Figure 6d*), each teacher model encodes a sequence (*Figure 7a*) in a session. Teacher models encode the goal location at learning rates $\gamma_{fc_{\text{normal}}}$ and $\gamma_{cf_{\text{normal}}}$. After wake learning, we simulate an extended period of sleep with 5000 replay periods in each model. We then present each teacher model's 5000 replayed sequences as inputs to train a different blank-slate student model with learning rates $\gamma_{replay}$. As described in *Figure 6d*, we computed the conditional probability that, following the reactivation of the $i$th item, the model immediately reactivates the item at position $i$ + lag, given that the $i$th item was available for reactivation.

# Additional information

## Funding

| Funder | Grant reference number | Author |
| --- | --- | --- |
| National Science Foundation | National Graduate Research Fellowship | Zhenglong Zhou |
| National Institute of Mental Health | R01 MH55687 | Michael J Kahana |

| Funder | Grant reference number | Author |
|---|---|---|
| National Institute of Mental Health | R01 MH129436 | Anna C Schapiro |

The funders had no role in study design, data collection and interpretation, or the decision to submit the work for publication.

### Author contributions

Zhenglong Zhou, Conceptualization, Software, Formal analysis, Validation, Investigation, Visualization, Methodology, Writing – original draft, Writing – review and editing; Michael J Kahana, Anna C Schapiro, Conceptualization, Formal analysis, Supervision, Methodology, Writing – original draft, Writing – review and editing

### Author ORCIDs
Zhenglong Zhou (iD) https://orcid.org/0000-0003-4003-7214
Michael J Kahana (iD) https://orcid.org/0000-0001-8122-9525
Anna C Schapiro (iD) https://orcid.org/0000-0001-8086-0331

Reviewer #1 (Public review): https://doi.org/10.7554/eLife.99931.3.sa1
Reviewer #3 (Public review): https://doi.org/10.7554/eLife.99931.3.sa2
Author response https://doi.org/10.7554/eLife.99931.3.sa3

## Additional files

### Supplementary files
MDAR checklist

### Data availability
This work is a computational modeling study, so no data were generated for this manuscript. The modeling code used to run all simulations in this study is available at https://github.com/schapirolab/CMR-replay (copy archived at *schapirolab, 2025*).

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
