## [Editor Report · eLife Assessment]

This is an **important** account of replay as recency-weighted context-guided memory reactivation that explains a number of empirical findings across human and rodent memory literatures. The evidence is **compelling** and the work is likely to inspire further adaptions to incorporate additional biological and cognitive features.

---

## [Referee Report · Reviewer #1 (Public review)]

Summary:

Zhou and colleagues developed a computational model of replay that heavily builds on cognitive models of memory in context (e.g., the context-maintenance and retrieval model), which have been successfully used to explain memory phenomena in the past. Their model produces results that mirror previous empirical findings in rodents and offers a new computational framework for thinking about replay.

Strengths:

The model is compelling and seems to explain a number of findings from the rodent literature. It is commendable that the authors implement commonly used algorithms from wakefulness to model sleep/rest, thereby linking wake and sleep phenomena in a parsimonious way. Additionally, the manuscript's comprehensive perspective on replay, bridging humans and non-human animals, enhanced its theoretical contribution.

Weaknesses:

This reviewer is not a computational neuroscientist by training, so some comments may stem from misunderstandings. I hope the authors would see those instances as opportunities to clarify their findings for broader audiences.

(1) The model predicts that temporally close items will be co-reactivated, yet evidence from humans suggests that temporal context doesn't guide sleep benefits (instead, semantic connections seem to be of more importance; Liu and Ranganath 2021, Schechtman et al 2023). Could these findings be reconciled with the model or is this a limitation of the current framework?

(2) During replay, the model is set so that the next reactivated item is sampled without replacement (i.e., the model cannot get "stuck" on a single item). I'm not sure what the biological backing behind this is and why the brain can't reactivate the same item consistently. Furthermore, I'm afraid that such a rule may artificially generate sequential reactivation of items regardless of wake training. Could the authors explain this better or show that this isn't the case?

(3) If I understand correctly, there are two ways in which novelty (i.e., less exposure) is accounted for in the model. The first and more talked about is the suppression mechanism (lines 639-646). The second is a change in learning rates (lines 593-595). It's unclear to me why both procedures are needed, how they differ, and whether these are two different mechanisms that the model implements. Also, since the authors controlled the extent to which each item was experienced during wakefulness, it's not entirely clear to me which of the simulations manipulated novelty on an individual item level, as described in lines 593-595 (if any).

As to the first mechanism - experience-based suppression - I find it challenging to think of a biological mechanism that would achieve this and is selectively activated immediately before sleep (somehow anticipating its onset). In fact, the prominent synaptic homeostasis hypothesis suggests that such suppression, at least on a synaptic level, is exactly what sleep itself does (i.e., prune or weaken synapses that were enhanced due to learning during the day). This begs the question of whether certain sleep stages (or ultradian cycles) may be involved in pruning, whereas others leverage its results for reactivation (e.g., a sequential hypothesis; Rasch & Born, 2013). That could be a compelling synthesis of this literature. Regardless of whether the authors agree, I believe that this point is a major caveat to the current model. It is addressed in the discussion, but perhaps it would be beneficial to explicitly state to what extent the results rely on the assumption of a pre-sleep suppression mechanism.

(4) As the manuscript mentions, the only difference between sleep and wake in the model is the initial conditions (a0). This is an obvious simplification, especially given the last author's recent models discussing the very different roles of REM vs NREM. Could the authors suggest how different sleep stages may relate to the model or how it could be developed to interact with other successful models such as the ones the last author has developed (e.g., C-HORSE)? Finally, I wonder how the model would explain findings (including the authors') showing a preference for reactivation of weaker memories. The literature seems to suggest that it isn't just a matter of novelty or exposure, but encoding strength. Can the model explain this? Or would it require additional assumptions or some mechanism for selective endogenous reactivation during sleep and rest?

(5) Lines 186-200 - Perhaps I'm misunderstanding, but wouldn't it be trivial that an external cue at the end-item of Figure 7a would result in backward replay, simply because there is no potential for forward replay for sequences starting at the last item (there simply aren't any subsequent items)? The opposite is true, of course, for the first-item replay, which can't go backward. More generally, my understanding of the literature on forward vs backward replay is that neither is linked to the rodent's location. Both commonly happen at a resting station that is further away from the track. It seems as though the model's result may not hold if replay occurs away from the track (i.e. if a0 would be equal for both pre- and post-run).

(6) The manuscript describes a study by Bendor & Wilson (2012) and tightly mimics their results. However, notably, that study did not find triggered replay immediately following sound presentation, but rather a general bias toward reactivation of the cued sequence over longer stretches of time. In other words, it seems that the model's results don't fully mirror the empirical results. One idea that came to mind is that perhaps it is the R/L context - not the first R/L item - that is cued in this study. This is in line with other TMR studies showing what may be seen as contextual reactivation. If the authors think that such a simulation may better mirror the empirical results, I encourage them to try. If not, however, this limitation should be discussed.

(7) There is some discussion about replay's benefit to memory. One point of interest could be whether this benefit changes between wake and sleep. Relatedly, it would be interesting to see whether the proportion of forward replay, backward replay, or both correlated with memory benefits. I encourage the authors to extend the section on the function of replay and explore these questions.

(8) Replay has been mostly studied in rodents, with few exceptions, whereas CMR and similar models have mostly been used in humans. Although replay is considered a good model of episodic memory, it is still limited due to limited findings of sequential replay in humans and its reliance on very structured and inherently autocorrelated items (i.e., place fields). I'm wondering if the authors could speak to the implications of those limitations on the generalizability of their model. Relatedly, I wonder if the model could or does lead to generalization to some extent in a way that would align with the complementary learning systems framework.

---

## [Referee Report · Reviewer #3 (Public review)]

In this manuscript, Zhou et al. present a computational model of memory replay. Their model (CMR-replay) draws from temporal context models of human memory (e.g., TCM, CMR) and claims replay may be another instance of a context-guided memory process. During awake learning, CMR-replay (like its predecessors) encodes items alongside a drifting mental context that maintains a recency-weighted history of recently encoded contexts/items. In this way, the presently encoded item becomes associated with other recently learned items via their shared context representation - giving rise to typical effects in recall such as primacy, recency and contiguity. Unlike its predecessors, CMR-replay has built in replay periods. These replay periods are designed to approximate sleep or wakeful quiescence, in which an item is spontaneously reactivated, causing a subsequent cascade of item-context reactivations that further update the model's items-context associations.

Using this model of replay, Zhou et al. were able to reproduce a variety of empirical findings in the replay literature: e.g., greater forward replay at the beginning of a track and more backwards replay at the end; more replay for rewarded events; the occurrence of remote replay; reduced replay for repeated items, etc. Furthermore, the model diverges considerably (in implementation and predictions) from other prominent models of replay that, instead, emphasize replay as a way of predicting value from a reinforcement learning framing (i.e., EVB, expected value backup).

Overall, I found the manuscript clear and easy to follow, despite not being a computational modeller myself. (Which is pretty commendable, I'd say). The model also was effective at capturing several important empirical results from the replay literature while relying on a concise set of mechanisms - which will have implications for subsequent theory building in the field.

The authors addressed my concerns with respect to adding methodological detail. I am satisfied with the changes.

---

## [Author Response]

The following is the authors’ response to the original reviews.

**Public Reviews:**

**Reviewer #1 (Public Review):**
Summary:Zhou and colleagues developed a computational model of replay that heavily builds on cognitive models of memory in context (e.g., the context-maintenance and retrieval model), which have been successfully used to explain memory phenomena in the past. Their model produces results that mirror previous empirical findings in rodents and offers a new computational framework for thinking about replay.Strengths:The model is compelling and seems to explain a number of findings from the rodent literature. It is commendable that the authors implement commonly used algorithms from wakefulness to model sleep/rest, thereby linking wake and sleep phenomena in a parsimonious way. Additionally, the manuscript's comprehensive perspective on replay, bridging humans and non-human animals, enhanced its theoretical contribution.Weaknesses:This reviewer is not a computational neuroscientist by training, so some comments may stem from misunderstandings. I hope the authors would see those instances as opportunities to clarify their findings for broader audiences.(1) The model predicts that temporally close items will be co-reactivated, yet evidence from humans suggests that temporal context doesn't guide sleep benefits (instead, semantic connections seem to be of more importance; Liu and Ranganath 2021, Schechtman et al 2023). Could these findings be reconciled with the model or is this a limitation of the current framework?

We appreciate the encouragement to discuss this connection. Our framework can accommodate semantic associations as determinants of sleep-dependent consolidation, which can in principle outweigh temporal associations. Indeed, prior models in this lineage have extensively simulated how semantic associations support encoding and retrieval alongside temporal associations. It would therefore be straightforward to extend our model to simulate how semantic associations guide sleep benefits, and to compare their contribution against that conferred by temporal associations across different experimental paradigms. In the revised manuscript, we have added a discussion of how our framework may simulate the role of semantic associations in sleep-dependent consolidation.

“Several recent studies have argued for dominance of semantic associations over temporal associations in the process of human sleep-dependent consolidation (Schechtman et al., 2023; Liu and Ranganath 2021; Sherman et al., 2025), with one study observing no role at all for temporal associations (Schechtman et al., 2023). At first glance, these findings appear in tension with our model, where temporal associations drive offline consolidation. Indeed, prior models have accounted for these findings by suppressing temporal context during sleep (Liu and Ranganath 2024; Sherman et al., 2025). However, earlier models in the CMR lineage have successfully captured the joint contributions of semantic and temporal associations to encoding and retrieval (Polyn et al., 2009), and these processes could extend naturally to offline replay. In a paradigm where semantic associations are especially salient during awake learning, the model could weight these associations more and account for greater co-reactivation and sleep-dependent memory benefits for semantically related than temporally related items. Consistent with this idea, Schechtman et al. (2023) speculated that their null temporal effects likely reflected the task’s emphasis on semantic associations. When temporal associations are more salient and task-relevant, sleep-related benefits for temporally contiguous items are more likely to emerge (e.g., Drosopoulos et al., 2007; King et al., 2017).”

The reviewer’s comment points to fruitful directions for future work that could employ our framework to dissect the relative contributions of semantic and temporal associations to memory consolidation.

(2) During replay, the model is set so that the next reactivated item is sampled without replacement (i.e., the model cannot get "stuck" on a single item). I'm not sure what the biological backing behind this is and why the brain can't reactivate the same item consistently.Furthermore, I'm afraid that such a rule may artificially generate sequential reactivation of items regardless of wake training. Could the authors explain this better or show that this isn't the case?

We appreciate the opportunity to clarify this aspect of the model. We first note that this mechanism has long been a fundamental component of this class of models (Howard & Kahana 2002). Many classic memory models (Brown et al., 2000; Burgess & Hitch, 1991; Lewandowsky & Murdock 1989) incorporate response suppression, in which activated items are temporarily inhibited. The simplest implementation, which we use here, removes activated items from the pool of candidate items. Alternative implementations achieve this through transient inhibition, often conceptualized as neuronal fatigue (Burgess & Hitch, 1991; Grossberg 1978). Our model adopts a similar perspective, interpreting this mechanism as mimicking a brief refractory period that renders reactivated neurons unlikely to fire again within a short physiological event such as a sharp-wave ripple. Importantly, this approach does not generate spurious sequences. Instead, the model’s ability to preserve the structure of wake experience during replay depends entirely on the learned associations between items (without these associations, item order would be random). Similar assumptions are also common in models of replay. For example, reinforcement learning models of replay incorporate mechanisms such as inhibition to prevent repeated reactivations (e.g., Diekmann & Cheng, 2023) or prioritize reactivation based on ranking to limit items to a single replay (e.g., Mattar & Daw, 2018). We now discuss these points in the section titled “A context model of memory replay”

“This mechanism of sampling without replacement, akin to response suppression in established context memory models (Howard & Kahana 2002), could be implemented by neuronal fatigue or refractory dynamics (Burgess & Hitch, 1991; Grossberg 1978). Non-repetition during reactivation is also a common assumption in replay models that regulate reactivation through inhibition or prioritization (Diekmann & Cheng 2023; Mattar & Daw 2018; Singh et al., 2022).”

(3) If I understand correctly, there are two ways in which novelty (i.e., less exposure) is accounted for in the model. The first and more talked about is the suppression mechanism (lines 639-646). The second is a change in learning rates (lines 593-595). It's unclear to me why both procedures are needed, how they differ, and whether these are two different mechanisms that the model implements. Also, since the authors controlled the extent to which each item was experienced during wakefulness, it's not entirely clear to me which of the simulations manipulated novelty on an individual item level, as described in lines 593-595 (if any).

We agree that these mechanisms and their relationships would benefit from clarification. As noted, novelty influences learning through two distinct mechanisms. First, the suppression mechanism is essential for capturing the inverse relationship between the amount of wake experience and the frequency of replay, as observed in several studies. This mechanism ensures that items with high wake activity are less likely to dominate replay. Second, the decrease in learning rates with repetition is crucial for preserving the stochasticity of replay. Without this mechanism, the model would increase weights linearly, leading to an exponential increase in the probability of successive wake items being reactivated back-to-back due to the use of a softmax choice rule. This would result in deterministic replay patterns, which are inconsistent with experimental observations.

We have revised the Methods section to explicitly distinguish these two mechanisms:

“This experience-dependent suppression mechanism is distinct from the reduction of learning rates through repetition; it does not modulate the update of memory associations but exclusively governs which items are most likely to initiate replay.”

We have also clarified our rationale for including a learning rate reduction mechanism:

“The reduction in learning rates with repetition is important for maintaining a degree of stochasticity in the model’s replay during task repetition, since linearly increasing weights would, through the softmax choice rule, exponentially amplify differences in item reactivation probabilities, sharply reducing variability in replay.”

Finally, we now specify exactly where the learning-rate reduction applied, namely in simulations where sequences are repeated across multiple sessions:

“In this simulation, the learning rates progressively decrease across sessions, as described above.“

As to the first mechanism - experience-based suppression - I find it challenging to think of a biological mechanism that would achieve this and is selectively activated immediately before sleep (somehow anticipating its onset). In fact, the prominent synaptic homeostasis hypothesis suggests that such suppression, at least on a synaptic level, is exactly what sleep itself does (i.e., prune or weaken synapses that were enhanced due to learning during the day). This begs the question of whether certain sleep stages (or ultradian cycles) may be involved in pruning, whereas others leverage its results for reactivation (e.g., a sequential hypothesis; Rasch & Born, 2013). That could be a compelling synthesis of this literature. Regardless of whether the authors agree, I believe that this point is a major caveat to the current model. It is addressed in the discussion, but perhaps it would be beneficial to explicitly state to what extent the results rely on the assumption of a pre-sleep suppression mechanism.

We appreciate the reviewer raising this important point. Unlike the mechanism proposed by the synaptic homeostasis hypothesis, the suppression mechanism in our model does not suppress items based on synapse strength, nor does it modify synaptic weights. Instead, it determines the level of suppression for each item based on activity during awake experience. The brain could implement such a mechanism by tagging each item according to its activity level during wakefulness. During subsequent consolidation, the initial reactivation of an item during replay would reflect this tag, influencing how easily it can be reactivated.

A related hypothesis has been proposed in recent work, suggesting that replay avoids recently active trajectories due to spike frequency adaptation in neurons (Mallory et al., 2024). Similarly, the suppression mechanism in our model is critical for explaining the observed negative relationship between the amount of recent wake experience and the degree of replay.

We discuss the biological plausibility of this mechanism and its relationship with existing models in the Introduction. In the section titled “The influence of experience”, we have added the following:

“Our model implements an activity‑dependent suppression mechanism that, at the onset of each offline replay event, assigns each item a selection probability inversely proportional to its activation during preceding wakefulness. The brain could implement this by tagging each memory trace in proportion to its recent activation; during consolidation, that tag would then regulate starting replay probability, making highly active items less likely to be reactivated. A recent paper found that replay avoids recently traversed trajectories through awake spike‑frequency adaptation (Mallory et al., 2025), which could implement this kind of mechanism. In our simulations, this suppression is essential for capturing the inverse relationship between replay frequency and prior experience. Note that, unlike the synaptic homeostasis hypothesis (Tononi & Cirelli 2006), which proposes that the brain globally downscales synaptic weights during sleep, this mechanism leaves synaptic weights unchanged and instead biases the selection process during replay.”

(4) As the manuscript mentions, the only difference between sleep and wake in the model is the initial conditions (a0). This is an obvious simplification, especially given the last author's recent models discussing the very different roles of REM vs NREM. Could the authors suggest how different sleep stages may relate to the model or how it could be developed to interact with other successful models such as the ones the last author has developed (e.g., C-HORSE)?

We appreciate the encouragement to comment on the roles of different sleep stages in the manuscript, especially since, as noted, the lab is very interested in this and has explored it in other work. We chose to focus on NREM in this work because the vast majority of electrophysiological studies of sleep replay have identified these events during NREM. In addition, our lab’s theory of the role of REM (Singh et al., 2022, PNAS) is that it is a time for the neocortex to replay remote memories, in complement to the more recent memories replayed during NREM. The experiments we simulate all involve recent memories. Indeed, our view is that part of the reason that there is so little data on REM replay may be that experimenters are almost always looking for traces of recent memories (for good practical and technical reasons).

Regarding the simplicity of the distinction between simulated wake and sleep replay, we view it as an asset of the model that it can account for many of the different characteristics of awake and NREM replay with very simple assumptions about differences in the initial conditions. There are of course many other differences between the states that could be relevant to the impact of replay, but the current target empirical data did not necessitate us taking those into account. This allows us to argue that differences in initial conditions should play a substantial role in an account of the differences between wake and sleep replay.

We have added discussion of these ideas and how they might be incorporated into future versions of the model in the Discussion section:

“Our current simulations have focused on NREM, since the vast majority of electrophysiological studies of sleep replay have identified replay events in this stage. We have proposed in other work that replay during REM sleep may provide a complementary role to NREM sleep, allowing neocortical areas to reinstate remote, already-consolidated memories that need to be integrated with the memories that were recently encoded in the hippocampus and replayed during NREM (Singh et al., 2022). An extension of our model could undertake this kind of continual learning setup, where the student but not teacher network retains remote memories, and the driver of replay alternates between hippocampus (NREM) and cortex (REM) over the course of a night of simulated sleep. Other differences between stages of sleep and between sleep and wake states are likely to become important for a full account of how replay impacts memory. Our current model parsimoniously explains a range of differences between awake and sleep replay by assuming simple differences in initial conditions, but we expect many more characteristics of these states (e.g., neural activity levels, oscillatory profiles, neurotransmitter levels, etc.) will be useful to incorporate in the future.”

Finally, I wonder how the model would explain findings (including the authors') showing a preference for reactivation of weaker memories. The literature seems to suggest that it isn't just a matter of novelty or exposure, but encoding strength. Can the model explain this? Or would it require additional assumptions or some mechanism for selective endogenous reactivation during sleep and rest?

We appreciate the encouragement to discuss this, as we do think the model could explain findings showing a preference for reactivation of weaker memories, as in Schapiro et al. (2018). In our framework, memory strength is reflected in the magnitude of each memory’s associated synaptic weights, so that stronger memories yield higher retrieved‑context activity during wake encoding than weaker ones. Because the model’s suppression mechanism reduces an item’s replay probability in proportion to its retrieved‑context activity, items with larger weights (strong memories) are more heavily suppressed at the onset of replay, while those with smaller weights (weaker memories) receive less suppression. When items have matched reward exposure, this dynamic would bias offline replay toward weaker memories, therefore preferentially reactivating weak memories.

In the section titled “The influence of experience”, we updated a sentence to discuss this idea more explicitly:

“Such a suppression mechanism may be adaptive, allowing replay to benefit not only the most recently or strongly encoded items but also to provide opportunities for the consolidation of weaker or older memories, consistent with empirical evidence (e.g., Schapiro et al. 2018; Yu et al., 2024).”

(5) Lines 186-200 - Perhaps I'm misunderstanding, but wouldn't it be trivial that an external cue at the end-item of Figure 7a would result in backward replay, simply because there is no potential for forward replay for sequences starting at the last item (there simply aren't any subsequent items)? The opposite is true, of course, for the first-item replay, which can't go backward. More generally, my understanding of the literature on forward vs backward replay is that neither is linked to the rodent's location. Both commonly happen at a resting station that is further away from the track. It seems as though the model's result may not hold if replay occurs away from the track (i.e. if a0 would be equal for both pre- and post-run).

In studies where animals run back and forth on a linear track, replay events are decoded separately for left and right runs, identifying both forward and reverse sequences for each direction, for example using direction-specific place cell sequence templates. Accordingly, in our simulation of, e.g., Ambrose et al. (2016), we use two independent sequences, one for left runs and one for right runs (an approach that has been taken in prior replay modeling work). Crucially, our model assumes a context reset between running episodes, preventing the final item of one traversal from acquiring contextual associations with the first item of the next. As a result, learning in the two sequences remains independent, and when an external cue is presented at the track’s end, replay predominantly unfolds in the backward direction, only occasionally producing forward segments when the cue briefly reactivates an earlier sequence item before proceeding forward.

We added a note to the section titled “The context-dependency of memory replay” to clarify this:

“In our model, these patterns are identical to those in our simulation of Ambrose et al. (2016), which uses two independent sequences to mimic the two run directions. This is because the drifting context resets before each run sequence is encoded, with the pause between runs acting as an event boundary that prevents the final item of one traversal from associating with the first item of the next, thereby keeping learning in each direction independent.”

To our knowledge, no study has observed a similar asymmetry when animals are fully removed from the track, although both types of replay can be observed when animals are away from the track. For example, Gupta et al. (2010) demonstrated that when animals replay trajectories far from their current location, the ratio of forward vs. backward replay appears more balanced. We now highlight this result in the manuscript and explain how it aligns with the predictions of our model:

“For example, in tasks where the goal is positioned in the middle of an arm rather than at its end, CMR-replay predicts a more balanced ratio of forward and reverse replay, whereas the EVB model still predicts a dominance of reverse replay due to backward gain propagation from the reward. This contrast aligns with empirical findings showing that when the goal is located in the middle of an arm, replay events are more evenly split between forward and reverse directions (Gupta et al., 2010), whereas placing the goal at the end of a track produces a stronger bias toward reverse replay (Diba & Buzsaki 2007).”

Although no studies, to our knowledge, have observed a context-dependent asymmetry between forward and backward replay when the animal is away from the track, our model does posit conditions under which it could. Specifically, it predicts that deliberation on a specific memory, such as during planning, could generate an internal context input that biases replay: actively recalling the first item of a sequence may favor forward replay, while thinking about the last item may promote backward replay, even when the individual is physically distant from the track.

We now discuss this prediction in the section titled “The context-dependency of memory replay”:

“Our model also predicts that deliberation on a specific memory, such as during planning, could serve to elicit an internal context cue that biases replay: actively recalling the first item of a sequence may favor forward replay, while thinking about the last item may promote backward replay, even when the individual is physically distant from the track. While not explored here, this mechanism presents a potential avenue for future modeling and empirical work.”

(6) The manuscript describes a study by Bendor & Wilson (2012) and tightly mimics their results. However, notably, that study did not find triggered replay immediately following sound presentation, but rather a general bias toward reactivation of the cued sequence over longer stretches of time. In other words, it seems that the model's results don't fully mirror the empirical results. One idea that came to mind is that perhaps it is the R/L context - not the first R/L item - that is cued in this study. This is in line with other TMR studies showing what may be seen as contextual reactivation. If the authors think that such a simulation may better mirror the empirical results, I encourage them to try. If not, however, this limitation should be discussed.

Although our model predicts that replay is triggered immediately by the sound cue, it also predicts a sustained bias toward the cued sequence. Replay in our model unfolds across the rest phase as multiple successive events, so the bias observed in our sleep simulations indeed reflects a prolonged preference for the cued sequence.

We now discuss this issue, acknowledging the discrepancy:

“Bendor and Wilson (2012) found that sound cues during sleep did not trigger immediate replay, but instead biased reactivation toward the cued sequence over an extended period of time. While the model does exhibit some replay triggered immediately by the cue, it also captures the sustained bias toward the cued sequence over an extended period.”

Second, within this framework, context is modeled as a weighted average of the features associated with items. As a result, cueing the model with the first R/L item produces qualitatively similar outcomes as cueing it with a more extended R/L cue that incorporates features of additional items. This is because both approaches ultimately use context features unique to the two sides.

(7) There is some discussion about replay's benefit to memory. One point of interest could be whether this benefit changes between wake and sleep. Relatedly, it would be interesting to see whether the proportion of forward replay, backward replay, or both correlated with memory benefits. I encourage the authors to extend the section on the function of replay and explore these questions.

We thank the reviewer for this suggestion. Regarding differences in the contribution of wake and sleep to memory, our current simulations predict that compared to rest in the task environment, sleep is less biased toward initiating replay at specific items, leading to a more uniform benefit across all memories. Regarding the contributions of forward and backward replay, our model predicts that both strengthen bidirectional associations between items and contexts, benefiting memory in qualitatively similar ways. Furthermore, we suggest that the offline learning captured by our teacher-student simulations reflects consolidation processes that are specific to sleep.

We have expanded the section titled **“**The influence of experience**”** to discuss these predictions of the model:

“The results outlined above arise from the model's assumption that replay strengthens bidirectional associations between items and contexts to benefit memory. This assumption leads to several predictions about differences across replay types. First, the model predicts that sleep yields different memory benefits compared to rest in the task environment: Sleep is less biased toward initiating replay at specific items, resulting in a more uniform benefit across all memories. Second, the model predicts that forward and backward replay contribute to memory in qualitatively similar ways but tend to benefit different memories. This divergence arises because forward and backward replay exhibit distinct item preferences, with backward replay being more likely to include rewarded items, thereby preferentially benefiting those memories.”

We also updated the “The function of replay” section to include our teacher-student speculation:

“We speculate that the offline learning observed in these simulations corresponds to consolidation processes that operate specifically during sleep, when hippocampal-neocortical dynamics are especially tightly coupled (Klinzing et al., 2019).”

(8) Replay has been mostly studied in rodents, with few exceptions, whereas CMR and similar models have mostly been used in humans. Although replay is considered a good model of episodic memory, it is still limited due to limited findings of sequential replay in humans and its reliance on very structured and inherently autocorrelated items (i.e., place fields). I'm wondering if the authors could speak to the implications of those limitations on the generalizability of their model. Relatedly, I wonder if the model could or does lead to generalization to some extent in a way that would align with the complementary learning systems framework.

We appreciate these insightful comments. Traditionally, replay studies have focused on spatial tasks with autocorrelated item representations (e.g., place fields). However, an increasing number of human studies have demonstrated sequential replay using stimuli with distinct, unrelated representations. Our model is designed to accommodate both scenarios. In our current simulations, we employ orthogonal item representations while leveraging a shared, temporally autocorrelated context to link successive items. We anticipate that incorporating autocorrelated item representations would further enhance sequence memory by increasing the similarity between successive contexts. Overall, we believe that the model generalizes across a broad range of experimental settings, regardless of the degree of autocorrelation between items. Moreover, the underlying framework has been successfully applied to explain sequential memory in both spatial domains, explaining place cell firing properties (e.g., Howard et al., 2004), and in non-spatial domains, such as free recall experiments where items are arbitrarily related.

In the section titled “A context model of memory replay”, we added this comment to address this point:

“Its contiguity bias stems from its use of shared, temporally autocorrelated context to link successive items, despite the orthogonal nature of individual item representations. This bias would be even stronger if items had overlapping representations, as observed in place fields.”

Since CMR-replay learns distributed context representations where overlap across context vectors captures associative structure, and replay helps strengthen that overlap, this could indeed be viewed as consonant with complementary learning systems integration processes.

**Reviewer #2 (Public Review):**
This manuscript proposes a model of replay that focuses on the relation between an item and its context, without considering the value of the item. The model simulates awake learning, awake replay, and sleep replay, and demonstrates parallels between memory phenomenon driven by encoding strength, replay of sequence learning, and activation of nearest neighbor to infer causality. There is some discussion of the importance of suppression/inhibition to reduce activation of only dominant memories to be replayed, potentially boosting memories that are weakly encoded. Very nice replications of several key replay findings including the effect of reward and remote replay, demonstrating the equally salient cue of context for offline memory consolidation.I have no suggestions for the main body of the study, including methods and simulations, as the work is comprehensive, transparent, and well-described. However, I would like to understand how the CMRreplay model fits with the current understanding of the importance of excitation vs inhibition, remembering vs forgetting, activation vs deactivation, strengthening vs elimination of synapses, and even NREM vs REM as Schapiro has modeled. There seems to be a strong association with the efforts of the model to instantiate a memory as well as how that reinstantiation changes across time. But that is not all this is to consolidation. The specific roles of different brain states and how they might change replay is also an important consideration.

We are gratified that the reviewer appreciated the work, and we agree that the paper would benefit from comment on the connections to these other features of consolidation.

Excitation vs. inhibition: CMR-replay does not model variations in the excitation-inhibition balance across brain states (as in other models, e.g., Chenkov et al., 2017), since it does not include inhibitory connections. However, we posit that the experience-dependent suppression mechanism in the model might, in the brain, involve inhibitory processes. Supporting this idea, studies have observed increased inhibition with task repetition (Berners-Lee et al., 2022). We hypothesize that such mechanisms may underlie the observed inverse relationship between task experience and replay frequency in many studies. We discuss this in the section titled “A context model of memory replay”:

“The proposal that a suppression mechanism plays a role in replay aligns with models that regulate place cell reactivation via inhibition (Malerba et al., 2016) and with empirical observations of increased hippocampal inhibitory interneuron activity with experience (Berners-Lee et al., 2022). Our model assumes the presence of such inhibitory mechanisms but does not explicitly model them.”

Remembering/forgetting, activation/deactivation, and strengthening/elimination of synapses: The model does not simulate synaptic weight reduction or pruning, so it does not forget memories through the weakening of associated weights. However, forgetting can occur when a memory is replayed less frequently than others, leading to reduced activation of that memory compared to its competitors during context-driven retrieval. In the Discussion section, we acknowledge that a biologically implausible aspect of our model is that it implements only synaptic strengthening:

“Aspects of the model, such as its lack of regulation of the cumulative positive weight changes that can accrue through repeated replay, are biologically implausible (as biological learning results in both increases and decreases in synaptic weights) and limit the ability to engage with certain forms of low level neural data (e.g., changes in spine density over sleep periods; de Vivo et al., 2017; Maret et al., 2011). It will be useful for future work to explore model variants with more elements of biological plausibility.” Different brain states and NREM vs REM: Reviewer 1 also raised this important issue (see above). We have added the following thoughts on differences between these states and the relationship to our prior work to the Discussion section:

“Our current simulations have focused on NREM, since the vast majority of electrophysiological studies of sleep replay have identified replay events in this stage. We have proposed in other work that replay during REM sleep may provide a complementary role to NREM sleep, allowing neocortical areas to reinstate remote, already-consolidated memories that need to be integrated with the memories that were recently encoded in the hippocampus and replayed during NREM (Singh et al., 2022). An extension of our model could undertake this kind of continual learning setup, where the student but not teacher network retains remote memories, and the driver of replay alternates between hippocampus (NREM) and cortex (REM) over the course of a night of simulated sleep. Other differences between stages of sleep and between sleep and wake states are likely to become important for a full account of how replay impacts memory. Our current model parsimoniously explains a range of differences between awake and sleep replay by assuming simple differences in initial conditions, but we expect many more characteristics of these states (e.g., neural activity levels, oscillatory profiles, neurotransmitter levels, etc.) will be useful to incorporate in the future.”

We hope these points clarify the model’s scope and its potential for future extensions.

Do the authors suggest that these replay systems are more universal to offline processes beyond episodic memory? What about procedural memories and working memory?

We thank the reviewer for raising this important question. We have clarified in the manuscript:

“We focus on the model as a formulation of hippocampal replay, capturing how the hippocampus may replay past experiences through simple and interpretable mechanisms.”

With respect to other forms of memory, we now note that:

“This motor memory simulation using a model of hippocampal replay is consistent with evidence that hippocampal replay can contribute to consolidating memories that are not hippocampally dependent at encoding (Schapiro et al., 2019; Sawangjit et al., 2018). It is possible that replay in other, more domain-specific areas could also contribute (Eichenlaub et al., 2020).”

Though this is not a biophysical model per se, can the authors speak to the neuromodulatory milieus that give rise to the different types of replay?

Our work aligns with the perspective proposed by Hasselmo (1999), which suggests that waking and sleep states differ in the degree to which hippocampal activity is driven by external inputs. Specifically, high acetylcholine levels during waking bias activity to flow into the hippocampus, while low acetylcholine levels during sleep allow hippocampal activity to influence other brain regions. Consistent with this view, our model posits that wake replay is more biased toward items associated with the current resting location due to the presence of external input during waking states. In the Discussion section, we have added a comment on this point:

“Our view aligns with the theory proposed by Hasselmo (1999), which suggests that the degree of hippocampal activity driven by external inputs differs between waking and sleep states: High acetylcholine levels during wakefulness bias activity into the hippocampus, while low acetylcholine levels during slow-wave sleep allow hippocampal activity to influence other brain regions.”

**Reviewer #3 (Public Review):**
In this manuscript, Zhou et al. present a computational model of memory replay. Their model (CMR-replay) draws from temporal context models of human memory (e.g., TCM, CMR) and claims replay may be another instance of a context-guided memory process. During awake learning, CMR replay (like its predecessors) encodes items alongside a drifting mental context that maintains a recency-weighted history of recently encoded contexts/items. In this way, the presently encoded item becomes associated with other recently learned items via their shared context representation - giving rise to typical effects in recall such as primacy, recency, and contiguity. Unlike its predecessors, CMR-replay has built-in replay periods. These replay periods are designed to approximate sleep or wakeful quiescence, in which an item is spontaneously reactivated, causing a subsequent cascade of item-context reactivations that further update the model's item-context associations.Using this model of replay, Zhou et al. were able to reproduce a variety of empirical findings in the replay literature: e.g., greater forward replay at the beginning of a track and more backward replay at the end; more replay for rewarded events; the occurrence of remote replay; reduced replay for repeated items, etc. Furthermore, the model diverges considerably (in implementation and predictions) from other prominent models of replay that, instead, emphasize replay as a way of predicting value from a reinforcement learning framing (i.e., EVB, expected value backup).Overall, I found the manuscript clear and easy to follow, despite not being a computational modeller myself. (Which is pretty commendable, I'd say). The model also was effective at capturing several important empirical results from the replay literature while relying on a concise set of mechanisms - which will have implications for subsequent theory-building in the field.With respect to weaknesses, additional details for some of the methods and results would help the readers better evaluate the data presented here (e.g., explicitly defining how the various 'proportion of replay' DVs were calculated).For example, for many of the simulations, the y-axis scale differs from the empirical data despite using comparable units, like the proportion of replay events (e.g., Figures 1B and C). Presumably, this was done to emphasize the similarity between the empirical and model data. But, as a reader, I often found myself doing the mental manipulation myself anyway to better evaluate how the model compared to the empirical data. Please consider using comparable y-axis ranges across empirical and simulated data wherever possible.

We appreciate this point. As in many replay modeling studies, our primary goal is to provide a qualitative fit that demonstrates the general direction of differences between our model and empirical data, without engaging in detailed parameter fitting for a precise quantitative fit. Still, we agree that where possible, it is useful to better match the axes. We have updated figures 2B and 2C so that the y-axis scales are more directly comparable between the empirical and simulated data.

In a similar vein to the above point, while the DVs in the simulations/empirical data made intuitive sense, I wasn't always sure precisely how they were calculated. Consider the "proportion of replay" in Figure 1A. In the Methods (perhaps under Task Simulations), it should specify exactly how this proportion was calculated (e.g., proportions of all replay events, both forwards and backwards, combining across all simulations from Pre- and Post-run rest periods). In many of the examples, the proportions seem to possibly sum to 1 (e.g., Figure 1A), but in other cases, this doesn't seem to be true (e.g., Figure 3A). More clarity here is critical to help readers evaluate these data. Furthermore, sometimes the labels themselves are not the most informative. For example, in Figure 1A, the y-axis is "Proportion of replay" and in 1C it is the "Proportion of events". I presumed those were the same thing - the proportion of replay events - but it would be best if the axis labels were consistent across figures in this manuscript when they reflect the same DV.

We appreciate these useful suggestions. We have revised the Methods section to explain in detail how DVs are calculated for each simulation. The revisions clarify the differences between related measures, such as those shown in Figures 1A and 1C, so that readers can more easily see how the DVs are defined and interpreted in each case.

**Reviewer #4/Reviewing Editor (Public Review):**
Summary:With their 'CMR-replay' model, Zhou et al. demonstrate that the use of spontaneous neural cascades in a context-maintenance and retrieval (CMR) model significantly expands the range of captured memory phenomena.Strengths:The proposed model compellingly outperforms its CMR predecessor and, thus, makes important strides towards understanding the empirical memory literature, as well as highlighting a cognitive function of replay.Weaknesses:Competing accounts of replay are acknowledged but there are no formal comparisons and only CMR-replay predictions are visualized. Indeed, other than the CMR model, only one alternative account is given serious consideration: A variant of the 'Dyna-replay' architecture, originally developed in the machine learning literature (Sutton, 1990; Moore & Atkeson, 1993) and modified by Mattar et al (2018) such that previously experienced event-sequences get replayed based on their relevance to future gain. Mattar et al acknowledged that a realistic Dyna-replay mechanism would require a learned representation of transitions between perceptual and motor events, i.e., a 'cognitive map'. While Zhou et al. note that the CMR-replay model might provide such a complementary mechanism, they emphasize that their account captures replay characteristics that Dyna-replay does not (though it is unclear to what extent the reverse is also true).

We thank the reviewer for these thoughtful comments and appreciate the opportunity to clarify our approach. Our goal in this work is to contrast two dominant perspectives in replay research: replay as a mechanism for learning reward predictions and replay as a process for memory consolidation. These models were chosen as representatives of their classes of models because they use simple and interpretable mechanisms that can simulate a wide range of replay phenomena, making them ideal for contrasting these two perspectives.

Although we implemented CMR-replay as a straightforward example of the memory-focused view, we believe the proposed mechanisms could be extended to other architectures, such as recurrent neural networks, to produce similar results. We now discuss this possibility in the revised manuscript (see below). However, given our primary goal of providing a broad and qualitative contrast of these two broad perspectives, we decided not to undertake simulations with additional individual models for this paper.

Regarding the Mattar & Daw model, it is true that a mechanistic implementation would require a mechanism that avoids precomputing priorities before replay. However, the "need" component of their model already incorporates learned expectations of transitions between actions and events. Thus, the model's limitations are not due to the absence of a cognitive map.

In contrast, while CMR-replay also accumulates memory associations that reflect experienced transitions among events, it generates several qualitatively distinct predictions compared to the Mattar & Daw model. As we note in the manuscript, these distinctions make CMR-replay a contrasting rather than complementary perspective.

Another important consideration, however, is how CMR replay compares to alternative mechanistic accounts of cognitive maps. For example, Recurrent Neural Networks are adept at detecting spatial and temporal dependencies in sequential input; these networks are being increasingly used to capture psychological and neuroscientific data (e.g., Zhang et al, 2020; Spoerer et al, 2020), including hippocampal replay specifically (Haga & Fukai, 2018). Another relevant framework is provided by Associative Learning Theory, in which bidirectional associations between static and transient stimulus elements are commonly used to explain contextual and cue-based phenomena, including associative retrieval of absent events (McLaren et al, 1989; Harris, 2006; Kokkola et al, 2019). Without proper integration with these modeling approaches, it is difficult to gauge the innovation and significance of CMR-replay, particularly since the model is applied post hoc to the relatively narrow domain of rodent maze navigation.

First, we would like to clarify our principal aim in this work is to characterize the nature of replay, rather than to model cognitive maps per se. Accordingly, CMR‑replay is not designed to simulate head‐direction signals, perform path integration, or explain the spatial firing properties of neurons during navigation. Instead, it focuses squarely on sequential replay phenomena, simulating classic rodent maze reactivation studies and human sequence‐learning tasks. These simulations span a broad array of replay experimental paradigms to ensure extensive coverage of the replay findings reported across the literature. As such, the contribution of this work is in explaining the mechanisms and functional roles of replay, and demonstrating that a model that employs simple and interpretable memory mechanisms not only explains replay phenomena traditionally interpreted through a value-based lens but also accounts for findings not addressed by other memory-focused models.

As the reviewer notes, CMR-replay shares features with other memory-focused models. However, to our knowledge, none of these related approaches have yet captured the full suite of empirical replay phenomena, suggesting the combination of mechanisms employed in CMR-replay is essential for explaining these phenomena. In the Discussion section, we now discuss the similarities between CMR-replay and related memory models and the possibility of integrating these approaches:

“Our theory builds on a lineage of memory-focused models, demonstrating the power of this perspective in explaining phenomena that have often been attributed to the optimization of value-based predictions. In this work, we focus on CMR-replay, which exemplifies the memory-centric approach through a set of simple and interpretable mechanisms that we believe are broadly applicable across memory domains. Elements of CMR-replay share similarities with other models that adopt a memory-focused perspective. The model learns distributed context representations whose overlaps encodes associations among items, echoing associative learning theories in which overlapping patterns capture stimulus similarity and learned associations (McLaren & Mackintosh 2002). Context evolves through bidirectional interactions between items and their contextual representations, mirroring the dynamics found in recurrent neural networks (Haga & Futai 2018; Levenstein et al., 2024). However, these related approaches have not been shown to account for the present set of replay findings and lack mechanisms—such as reward-modulated encoding and experience-dependent suppression—that our simulations suggest are essential for capturing these phenomena. While not explored here, we believe these mechanisms could be integrated into architectures like recurrent neural networks (Levenstein et al., 2024) to support a broader range of replay dynamics.”

**Recommendations For The Authors**

**Reviewer #1 (Recommendations For The Authors):**
(1) Lines 94-96: These lines may be better positioned earlier in the paragraph.

We now introduce these lines earlier in the paragraph.

(2) Line 103 - It's unclear to me what is meant by the statement that "the current context contains contexts associated with previous items". I understand why a slowly drifting context will coincide and therefore link with multiple items that progress rapidly in time, so multiple items will be linked to the same context and each item will be linked to multiple contexts. Is that the idea conveyed here or am I missing something? I'm similarly confused by line 129, which mentions that a context is updated by incorporating other items' contexts. How could a context contain other contexts?

In the model, each item has an associated context that can be retrieved via Mfc. This is true even before learning, since Mfc is initialized as an identity matrix. During learning and replay, we have a drifting context c that is updated each time an item is presented. At each timestep, the model first retrieves the current item’s associated context cf by Mfc, and incorporates it into c. Equation #2 in the Methods section illustrates this procedure in detail. Because of this procedure, the drifting context c is a weighted sum of past items’ associated contexts.

We recognize that these descriptions can be confusing. We have updated the Results section to better distinguish the drifting context from items’ associated context. For example, we note that:

“We represent the drifting context during learning and replay with c and an item's associated context with cf.”

We have also updated our description of the context drift procedure to distinguish these two quantities:

“During awake encoding of a sequence of items, for each item f, the model retrieves its associated context cf via Mfc. The drifting context c incorporates the item's associated context cf and downweights its representation of previous items' associated contexts (Figure 1c). Thus, the context layer maintains a recency weighted sum of past and present items' associated contexts.”

(3) Figure 1b and 1d - please clarify which axis in the association matrices represents the item and the context.

We have added labels to show what the axes represent in Figure 1.

(4) The terms "experience" and "item" are used interchangeably and it may be best to stick to one term.

We now use the term “item” wherever we describe the model results.

(5) The manuscript describes Figure 6 ahead of earlier figures - the authors may want to reorder their figures to improve readability.

We appreciate this suggestion. We decided to keep the current figure organization since it allows us to group results into different themes and avoid redundancy.

(6) Lines 662-664 are repeated with a different ending, this is likely an error.

We have fixed this error.

**Reviewer #3 (Recommendations For The Authors):**
Below, I have outlined some additional points that came to mind in reviewing the manuscript - in no particular order.(1) Figure 1: I found the ordering of panels a bit confusing in this figure, as the reading direction changes a couple of times in going from A to F. Would perhaps putting panel C in the bottom left corner and then D at the top right, with E and F below (also on the right) work?

We agree that this improves the figure. We have restructured the ordering of panels in this figure.

(2) Simulation 1: When reading the intro/results for the first simulation (Figure 2a; Diba & Buszaki, 2007; "When animals traverse a linear track...", page 6, line 186). It wasn't clear to me why pre-run rest would have any forward replay, particularly if pre-run implied that the animal had no experience with the track yet. But in the Methods this becomes clearer, as the model encodes the track eight times prior to the rest periods. Making this explicit in the text would make it easier to follow. Also, was there any reason why specifically eight sessions of awake learning, in particular, were used?

We now make more explicit that the animals have experience with the track before pre-run rest recording:

“Animals first acquire experience with a linear track by traversing it to collect a reward. Then, during the pre-run rest recording, forward replay predominates.”

We included eight sessions of awake learning to match with the number of sessions in Shin et al. (2017), since this simulation attempts to explain data from that study. After each repetition, the model engages in rest. We have revised the Methods section to indicate the motivation for this choice:

“In the simulation that examines context-dependent forward and backward replay through experience (Figs. 2a and 5a), CMR-replay encodes an input sequence shown in Fig. 7a, which simulates a linear track run with no ambiguity in the direction of inputs, over eight awake learning sessions (as in Shin et al. 2019)”

(3) Frequency of remote replay events: In the simulation based on Gupta et al, how frequently overall does remote replay occur? In the main text, the authors mention the mean frequency with which shortcut replay occurs (i.e., the mean proportion of replay events that contain a shortcut sequence = 0.0046), which was helpful. But, it also made me wonder about the likelihood of remote replay events. I would imagine that remote replay events are infrequent as well - given that it is considerably more likely to replay sequences from the local track, given the recency-weighted mental context. Reporting the above mean proportion for remote and local replay events would be helpful context for the reader.

In Figure 4c, we report the proportion of remote replay in the two experimental conditions of Gupta et al. that we simulate.

(4) Point of clarification re: backwards replay: Is backwards replay less likely to occur than forward replay overall because of the forward asymmetry associated with these models? For example, for a backwards replay event to occur, the context would need to drift backwards at least five times in a row, in spite of a higher probability of moving one step forward at each of those steps. Am I getting that right?

The reviewer’s interpretation is correct: CMR-replay is more likely to produce forward than backward replay in sleep because of its forward asymmetry. We note that this forward asymmetry leads to high likelihood of forward replay in the section titled “The context-dependency of memory replay”:

“As with prior retrieved context models (Howard & Kahana 2002; Polyn et al., 2009), CMR-replay encodes stronger forward than backward associations. This asymmetry exists because, during the first encoding of a sequence, an item's associated context contributes only to its ensuing items' encoding contexts. Therefore, after encoding, bringing back an item's associated context is more likely to reactivate its ensuing than preceding items, leading to forward asymmetric replay (Fig. 6d left).”

(5) On terminating a replay period: "At any t, the replay period ends with a probability of 0.1 or if a task-irrelevant item is reactivated." (Figure 1 caption; see also pg 18, line 635). How was the 0.1 decided upon? Also, could you please add some detail as to what a 'task-irrelevant item' would be? From what I understood, the model only learns sequences that represent the points in a track - wouldn't all the points in the track be task-relevant?

This value was arbitrarily chosen as a small value that allows probabilistic stopping. It was not motivated by prior modeling or a systematic search. We have added: “At each timestep, the replay period ends either with a stop probability of 0.1 or if a task-irrelevant item becomes reactivated. (The choice of the value 0.1 was arbitrary; future work could explore the implications of varying this parameter).”

In addition, we now explain in the paper that task irrelevant items “do not appear as inputs during awake encoding, but compete with task-relevant items for reactivation during replay, simulating the idea that other experiences likely compete with current experiences during periods of retrieval and reactivation.”

(6) Minor typos:Turn all instances of "nonlocal" into "non-local", or vice versa"For rest at the end of a run, cexternal is the context associated with the final item in the sequence. For rest at the end of a run, cexternal is the context associated with the start item." (pg 20, line 663) - I believe this is a typo and that the second sentence should begin with "For rest at the START of a run".

We have updated the manuscript to correct these typos.

(7) Code availability: I may have missed it, but it doesn't seem like the code is currently available for these simulations. Including the commented code in a public repository (Github, OSF) would be very useful in this case.

We now include a Github link to our simulation code: https://github.com/schapirolab/CMR-replay.